# Coupled Distributional Random Expert Distillation for World Model Online Imitation Learning

## Abstract

Imitation Learning (IL) has achieved remarkable success across various domains, including robotics, autonomous driving, and healthcare, by enabling agents to learn complex behaviors from expert demonstrations. However, existing IL methods often face instability challenges, particularly when relying on adversarial reward or value formulations in world model frameworks. In this work, we propose a novel approach to online imitation learning that addresses these limitations through a reward model based on random network distillation (RND) for density estimation. Our reward model is built on the joint estimation of expert and behavioral distributions within the latent space of the world model. We evaluate our method across diverse benchmarks, including DMControl, Meta-World, and ManiSkill2, showcasing its ability to deliver stable performance and achieve expert-level results in both locomotion and manipulation tasks. Our approach demonstrates improved stability over adversarial methods while maintaining expert-level performance.

## 1 Introduction

Imitation Learning (IL) has recently shown remarkable effectiveness across a wide range of domains, particularly in addressing complex real-world challenges. In robotics, IL has significantly advanced the state of the art in manipulation (Zhu et al., 2022; Wan et al., 2024; Stepputtis et al., 2020; Chi et al., 2023) and locomotion tasks (Chiu et al., 2024; Seo et al., 2023; Huang et al., 2024), where it has facilitated the development of robust controllers for various robotic platforms. Beyond robotics, IL has also demonstrated its versatility in domains such as autonomous driving (Pan et al., 2017; Bronstein et al., 2022; Cheng et al., 2024), where it is used to model complex decision-making processes and ensure safe and efficient vehicle navigation. Moreover, IL has started making meaningful contributions to healthcare (Deuschel et al., 2023), providing support in medical decision-making and enhancing the interpretability of complex diagnostic processes. These achievements highlight the broad applicability of IL and its potential to drive transformative progress across diverse fields.

The simplest approach to imitation learning is to apply behavioral cloning directly to the provided expert dataset, as demonstrated in prior works like IBC (Florence et al., 2022) and Diffusion Policy (Chi et al., 2023). However, this approach is not dynamics aware and may result in lack of generalization when encountering out-of-distribution states. To address these shortcomings, methods like GAIL (Ho & Ermon, 2016), SQIL (Reddy et al., 2019), IQ-Learn (Garg et al., 2021), MAIL (Baram et al., 2016) and CFIL (Freund et al., 2023) have introduced value or reward estimation to facilitate a deeper understanding of the environment, while leveraging online interactions to enhance exploration. Specifically, GAIL, MAIL, and IQ-Learn frame the imitation learning problem as an adversarial training process, distinguishing between the state-action distributions of the expert and the learner.

Recent advancements in latent world models for imitation learning have made significant progress. Several prior works, including V-MAIL (Rafailov et al., 2021), CMIL (Kolev et al., 2024), Ditto (DeMoss et al., 2023), EfficientImitate (Yin et al., 2022), and IQ-MPC (Li et al., 2024), have integrated adversarial imitation learning frameworks with world models to address imitation learning tasks. However, as discussed in the experiment section, we found that even with world models, the

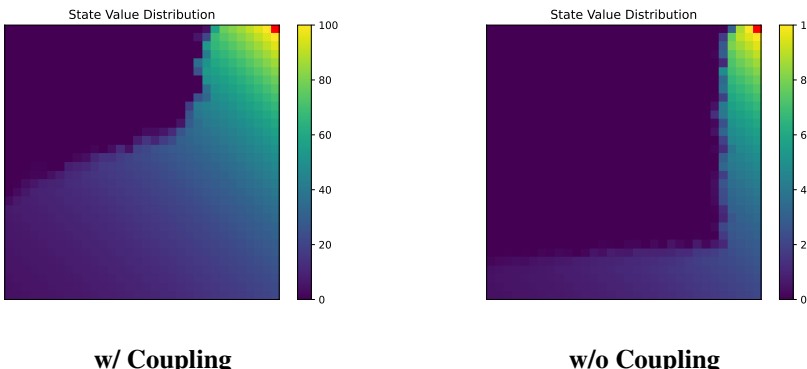

**w/ Coupling**    **w/o Coupling**

Figure 1: **Toy Example for Coupled Distribution Estimation** We present a toy experiment on a $32 \times 32$ GridWorld environment, comparing learning outcomes with and without a coupled reward estimator. Using only a single expert trajectory, we estimate rewards and perform Q-learning, as the environment has discrete state and action spaces. The empirical results show that incorporating coupled reward estimation significantly increases the state coverage, compared to estimating rewards solely from the expert trajectory. This highlights the coupled reward model's ability to encourage broader exploration.

adversarial objectives can still suffer from instability in certain scenarios. To overcome this issue, we propose replacing the adversarial reward or value formulation with a novel density estimation approach based on random network distillation (RND) (Burda et al., 2018), which mitigates the instability. Specifically, we perform density estimation in the latent space of the world model, leveraging the superior properties of latent representations and their enhanced dynamics-awareness, as the latent dynamics model is trained directly within this space. Unlike existing methods that use RND for imitation learning (Wang et al., 2019), our approach jointly learns the reward model and other components of the world model, estimating both the expert and behavioral distributions simultaneously in the latent space of the world model. In contrast, the existing Random Expert Distillation (Wang et al., 2019) estimates distributions in the original observation and action spaces, decouples the reward model learning from the downstream RL process, and does not include a coupled estimation on both expert and behavioral distributions, making it hard to solve complex tasks with high dimensional observation and action spaces. To demonstrate the effectiveness of our approach, we conduct evaluation across a range of tasks in DMControl (Tassa et al., 2018), Meta-World (Yu et al., 2020a), and ManiSkill2 (Gu et al., 2023), demonstrating stable performance and achieving expert-level results.

In conclusion, the contributions of our work are summarized as follows:

- We propose a novel reward model formulation for world model online imitation learning based on a coupled density estimation in the latent space of the world model.

- We demonstrate that our approach exhibits superior stability compared to previous approaches with adversarial formulations and achieves expert-level performance across a range of imitation learning tasks, including both locomotion and manipulation.

## 2 PRELIMINARY

We formulate our decision-making problem as Markov Decision Processes (MDPs). MDPs can be defined via a tuple $\langle \mathcal{S}, \mathcal{A}, p_0, \mathcal{P}, r, \gamma \rangle$. In details, $\mathcal{S}$ and $\mathcal{A}$ represent the state and action spaces, $p_0$ is the initial state distribution, $\mathcal{P} : \mathcal{S} \times \mathcal{A} \to \Delta_{\mathcal{S}}$ depicts the transition probability, $r(\mathbf{s}, \mathbf{a})$ is the reward function, and $\gamma \in (0, 1)$ is the discount factor. Let $\mathcal{Z}$ denote the latent state space of the world model. The expert latent state-action distribution and the behavioral latent state-action distribution (induced by the behavioral policy $\pi$) over $\mathcal{Z} \times \mathcal{A}$ are denoted by $\rho_E$ and $\rho_\pi$, respectively.

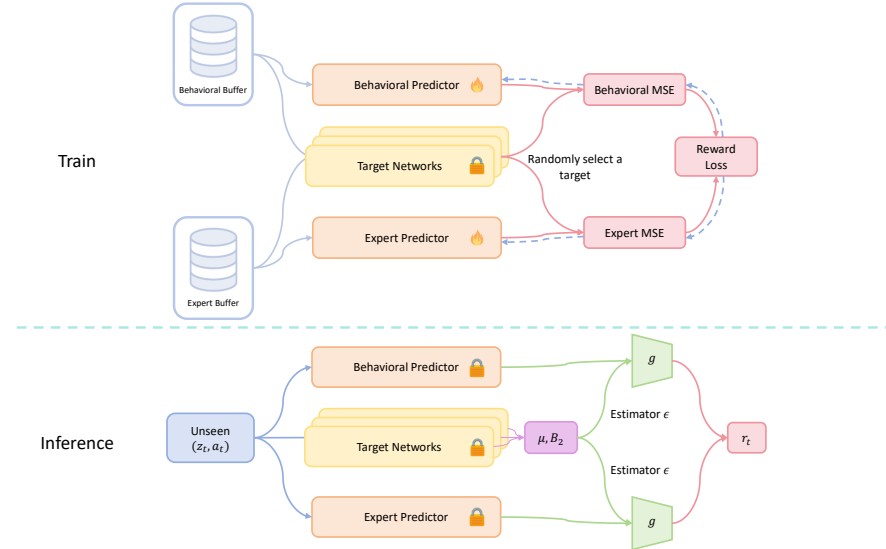

Figure 2: **Coupled Distributional Random Expert Distillation** We present the architecture of our CDRED reward model. During training, the behavioral and expert predictors are trained using latent representations encoded from observations and actions sampled from the behavioral and expert buffers. The dotted blue lines indicate the gradient backpropagation paths. During inference, rewards are estimated by the outputs of the behavioral and expert predictors, along with the mean and second-order moments of the target network's output, for an unseen latent state-action pair.

## 2.1 RANDOM NETWORK DISTILLATION FOR IMITATION LEARNING

Random Network Distillation (RND) (Burda et al., 2018) is a technique for promoting exploration. In details, it leverages a fixed randomly parameterized network $f_{\bar{\theta}}(x)$ and a learnable predictor network $f_\theta(x)$. During training, RND minimizes the following MSE loss for dataset $\mathcal{D}$ for certain data distribution $\rho$:

$$\mathcal{L}_{RND}(\theta) = \mathbb{E}_{x \sim \mathcal{D}} \|f_{\bar{\theta}}(x) - f_\theta(x)\|_2^2 \tag{1}$$

During the evaluation, we obtain a data point $x'$ for unknown data distribution $\rho'$. By computing the L2 norm $\|f_{\bar{\theta}}(x') - f_\theta(x')\|_2^2$, we can estimate the difference between distribution $\rho$ and $\rho'$. This can also be interpreted as performing density estimation for the new data point $x'$ within the original distribution $\rho$. A similar methodology has been used in imitation learning and inverse reinforcement learning, called Random Expert Distillation (Wang et al., 2019). In details, this approach performs imitation learning by estimating the support of expert policy distribution. During training, it minimizes $K$ pairs of predictors and fixed random targets in expert dataset with $N$ data points $\mathcal{D}_E = \{\mathbf{s}_i, \mathbf{a}_i\}_{0:N}$:

$$\hat{\theta}_k = \operatorname*{argmin}_\theta \frac{1}{N} \sum_{i=0}^{N-1} (f_\theta(\mathbf{s}_i, \mathbf{a}_i) - f_{\bar{\theta}_k}(\mathbf{s}_i, \mathbf{a}_i))^2 \tag{2}$$

In order to determine if a state-action pair is within the support of expert policy, it computes the L2 norm deviation for an unknown state-action pair $(\mathbf{s}, \mathbf{a})$ using $K$ pairs of predictors and targets:

$$\mathcal{L}_{RED}(\mathbf{s}, \mathbf{a}) = \frac{1}{K} \sum_{k=0}^{K-1} (f_{\hat{\theta}_k}(\mathbf{s}, \mathbf{a}) - f_{\bar{\theta}_k}(\mathbf{s}, \mathbf{a}))^2 \tag{3}$$

By leveraging a reward in the shape of $r(\mathbf{s}, \mathbf{a}) = \exp(-\sigma \, \mathcal{L}_{RED}(\mathbf{s}, \mathbf{a}))$, the approach effectively guides the downstream RL policy towards the expert distribution. However, this method may encounter challenges when the initial behavioral policy distribution is far from the expert distribution or when RED is applied naively on large latent spaces in world models.

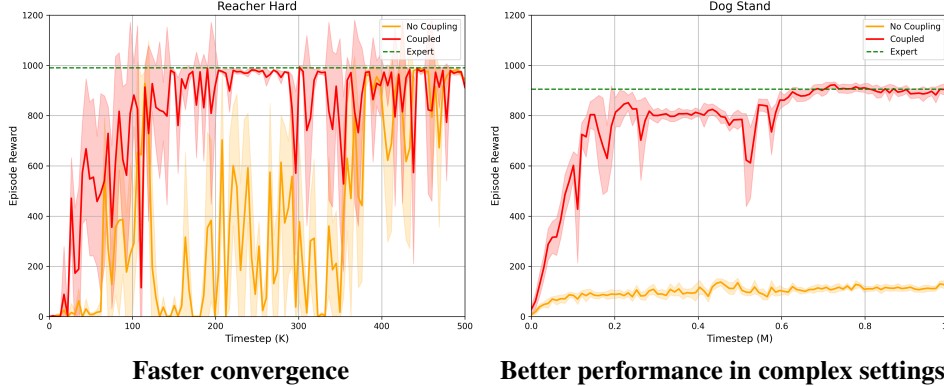

**Faster convergence**                    **Better performance in complex settings**

Figure 3: **Advantages of Coupled Density Estimation** We demonstrate the empirical performance boost of our coupled density estimation in terms of leveraging random network distillations for reward modeling based on state-action distribution estimation. With coupled estimation, we observe faster convergence to optimal in many simple cases (Left) and better performance in complex tasks (Right).

## 2.2 WORLD MODELS

Recent world models in the context of robotics control and reinforcement learning often represent a model-based RL method with latent spaces. The model learns a latent state transition model $\mathbf{z}' = d_\theta(\mathbf{z}, \mathbf{a})$, along with a encoder $\mathbf{z} = h_\theta(\mathbf{z})$ and a policy model $\mathbf{a} = \pi_\theta(\mathbf{z})$. The decision-making process often includes planning with latent unrolling. For models based on the Recurrent State-space Model (RSSM) (Hafner et al., 2019b), the latent states often are split into a deterministic part and a stochastic part. PlaNet (Hafner et al., 2019a) and Dreamer series (Hafner et al., 2019b; 2020; 2023) leverage decoders for observation reconstruction, while TD-MPC series (Hansen et al., 2022; 2023) leverages a decoder-free architecture and conducts planning solely in the latent space.

## 3 METHODOLOGY

In this section, we will go over the motivation and detailed methodology of our method, **C**oupled **D**istributional **R**andom **E**xpert **D**istillation, or **CDRED** as an abbreviation. We show that our method is stabler and more reasonable compared to naively apply Random Expert Distillation (RED) (Wang et al., 2019) on imitation learning with world models. To address the difficulties posed in RED as discussed in Section 2.1, we introduce a coupled distribution estimation approach in the latent space. This approach jointly estimates both the expert distribution and the behavioral distribution; it encourages policy exploration during the early stages of training. We present a toy example in Figure 1 to illustrate how coupled estimation promotes exploration, and provide the detailed methodology in Section 3.2. In this coupled approach, we need to estimate the behavioral distribution during online training, which naturally raises the problem of inconsistent final rewards, as noted by Yang et al. (2024). Thus, we adopt their method for tracking the frequency of data occurrence, which we describe in Section 3.1.

### 3.1 MITIGATING INCONSISTENT REWARD ESTIMATION

Inconsistencies can arise at various stages of RND model training (Yang et al., 2024). During the initial stage, these inconsistencies stem from extreme values in one network, which can be mitigated by using multiple target networks (denoted as $K$ target networks). In the final stage, inconsistencies occur when the resulting reward distribution does not align with the actual state-action distribution. To address this, an unbiased estimator for the state-action occurrence count $n$ is necessary. We should track state-action occurrence frequencies in order to maintain consistency when the distributional RND model is trained online. In this section, we replace the original state $\mathbf{s}_t$ with the latent representation $\mathbf{z}_t$ for the input of the RND model. Following Yang et al. (2024), we denote the random variable $c(\mathbf{z}_t, \mathbf{a}_t)$ as the output of a target network $f_{\bar{\theta}_k}$, where $k$ is sampled uniformly from the

interval $[0, K)$. For a predictor $f$ estimating a distribution $\rho$ (which can be either the expert distribution $\rho_E$ or the behavioral distribution $\rho_\pi$), by minimizing the $L_2$-norm loss $\|f(\mathbf{z}_t, \mathbf{a}_t) - c(\mathbf{z}_t, \mathbf{a}_t)\|_2^2$, the optimal predictor $f^*(\mathbf{z}_t, \mathbf{a}_t)$ is given by:

$$f^*(\mathbf{z}_t, \mathbf{a}_t) = \frac{1}{n} \sum_{i=1}^{n} c_i(\mathbf{z}_t, \mathbf{a}_t) \tag{4}$$

where $c_i(\mathbf{z}_t, \mathbf{a}_t)$ is representing the $c(\mathbf{z}_t, \mathbf{a}_t)$ for the $i$-th occurrence for state-action pair $(\mathbf{z}_t, \mathbf{a}_t)$ in distribution $\rho$. In order to track the occurrence count $n$, we adopt a lemma proposed by Yang et al. (2024):

**Lemma 1** (Unbiased Estimator). *For a state-action distribution $\rho$, $f^*$ is the optimal predictor on this distribution defined in Eq. 4, the following statistic is an unbiased estimator of $1/n$ with consistency for this distribution:*

$$y(\mathbf{z}_t, \mathbf{a}_t) = \frac{[f^*(\mathbf{z}_t, \mathbf{a}_t)]^2 - [\mu_{\bar{\theta}}(\mathbf{z}_t, \mathbf{a}_t)]^2}{B_2(\mathbf{z}_t, \mathbf{a}_t) - [\mu_{\bar{\theta}}(\mathbf{z}_t, \mathbf{a}_t)]^2}$$

*where the second-order moment is:*

$$B_2(\mathbf{z}_t, \mathbf{a}_t) = \frac{1}{K} \sum_{k=0}^{K-1} [f_{\bar{\theta}_k}(\mathbf{z}_t, \mathbf{a}_t)]^2$$

*Proof.* See Appendix F or prior work (Yang et al., 2024). $\square$

In this way, we are able to estimate the data distribution with higher consistency as the training proceeds. Following Yang et al. (2024), we construct the following estimator for $\sqrt{1/n}$ as an additional bonus correction term:

$$\epsilon(\mathbf{z}_t, \mathbf{a}_t, f) = \sqrt{\frac{[f(\mathbf{z}_t, \mathbf{a}_t)]^2 - [\mu_{\bar{\theta}}(\mathbf{z}_t, \mathbf{a}_t)]^2}{B_2(\mathbf{z}_t, \mathbf{a}_t) - [\mu_{\bar{\theta}}(\mathbf{z}_t, \mathbf{a}_t)]^2}} \tag{5}$$

This bonus correction is incorporated into the reward model construction discussed in Section 3.2.

## 3.2 COUPLED DISTRIBUTIONAL RANDOM EXPERT DISTILLATION

We construct a reward model with two predictor networks that share the same random target ensemble on the latent space of a world model. The distributional random target ensemble consists of $K$ random networks $\{f_{\bar{\theta}_k}\}_{0:K}$ with fixed parameters. Regarding the predictors, one of them is the expert predictor $f_\phi$ while the other is the behavioral predictor $f_\psi$. A predictor $f$ is defined by $f : \mathcal{Z} \times \mathcal{A} \to \mathbb{R}^p$, while $p$ is the dimension of the low-dimensional embedding space for L2 norm distance computation. Following Yang et al. (2024), we ask these two predictors to learn the random targets sampled. This is different to RED which learn $K$ predictors for $K$ targets. Given an expert buffer $\mathcal{B}_E$ and a behavioral buffer $\mathcal{B}_\pi$, we aim to optimize through the following objective:

$$\mathcal{L}^r(\phi, \psi) = \sum_{t=0}^{H} \lambda^t \, \mathbb{E}_{k \sim \text{Uniform}(0, K)} \left[ \mathbb{E}_{(\mathbf{s}_t, \mathbf{a}_t) \sim \mathcal{B}_E} \left[ \|f_\phi(\mathbf{z}_t, \mathbf{a}_t) - f_{\bar{\theta}_k}(\mathbf{z}_t, \mathbf{a}_t)\|_2^2 \right] \right.$$
$$\left. + \mathbb{E}_{(\mathbf{s}_t, \mathbf{a}_t) \sim \mathcal{B}_\pi} \left[ \|f_\psi(\mathbf{z}_t, \mathbf{a}_t) - f_{\bar{\theta}_k}(\mathbf{z}_t, \mathbf{a}_t)\|_2^2 \right] \right] \tag{6}$$

We sample short trajectories with horizon $H$ from the replay buffers and sum up the loss for every step with a discounting factor $\lambda$. Note that this factor is different from the environment discount factor $\gamma$. We update every time with one target network $f_{\bar{\theta}_k}$, where index $k$ is sampled from a uniform distribution over integers ranging $[0, K)$. $\mathbf{z}_t$ is the latent representation of $\mathbf{s}_t$ with an encoder mapping $\mathbf{z}_t = h(\mathbf{s}_t)$. In this way, we can obtain the estimation for expert distribution $\rho_E$ and behavioral distribution $\rho_\pi$. Furthermore, it enables us to construct a reward model based on the

distribution estimations. Incorporating the bias correction term introduced in Eq. 5, we are able to compute the reward via:

$$R(\mathbf{z}_t, \mathbf{a}_t) = \zeta \, g(-\sigma \, b(\mathbf{z}_t, \mathbf{a}_t, f_\phi)) - (1 - \zeta) \, g(-\sigma \, b(\mathbf{z}_t, \mathbf{a}_t, f_\psi)) \tag{7}$$

where

$$b(\mathbf{z}_t, \mathbf{a}_t, f) = \alpha \, \|f(\mathbf{z}_t, \mathbf{a}_t) - \mu_{\bar{\theta}}(\mathbf{z}_t, \mathbf{a}_t)\|_2^2 + (1 - \alpha) \, \epsilon(\mathbf{z}_t, \mathbf{a}_t, f) \tag{8}$$

$$\mu_{\bar{\theta}}(\mathbf{z}_t, \mathbf{a}_t) = \frac{1}{K} \sum_{k=0}^{K-1} f_{\bar{\theta}_k}(\mathbf{z}_t, \mathbf{a}_t) \tag{9}$$

The first term in Eq. 7 measures the distance between the current and expert distributions, while the second term encourages exploration by penalizing exploitation. A scaling factor $\zeta$ balances these terms, with the second term dominating during early training when the policy is sub-optimal, promoting exploration. As the policy approaches optimality, the first term takes over, stabilizing the policy near the expert distribution. Typically, $\zeta$ is close to 1, allowing the first term to dominate after initial exploration. The coefficient $\sigma$ controls the decay rate of the reward function, which is based on the expert distribution for the first term and the behavioral distribution for the second. To ensure stability, the reward is computed using the mean output of $K$ random target networks. The function $g(x)$ is monotonically increasing, and both $g(x) = \exp(x)$ and $g(x) = x$ work, with slight differences in behavior, as discussed in Appendix E.2. The scalar coefficient $\alpha$ in Eq. 8 balances the contributions of the first term (the $L_2$-norm) and the second term (an estimator for $\sqrt{1/n}$). Following Yang et al. (2024), we let the first term dominate initially, switching to the second term as training progresses. This can be achieved with a fixed $\alpha$, rather than a dynamic coefficient. This modification enables consistent online estimation of the state-action distribution, directly supporting reward modeling for online imitation learning.

### 3.3 INTEGRATING INTO WORLD MODELS FOR IMITATION LEARNING

World models learn the policy and underlying environment dynamics by encoding the observations into a latent space and learning the transition model in the latent space. Decoder-free world models such as TD-MPC series (Hansen et al., 2022; 2023) has proved to be a powerful tool for complex reinforcement learning tasks. We leverage a decoder-free world model containing the following components:

$$\text{Encoder:} \quad \mathbf{z}_t = h(\mathbf{s}_t) \tag{10}$$

$$\text{Latent dynamics:} \quad \mathbf{z}'_t = d(\mathbf{z}_t, \mathbf{a}_t) \tag{11}$$

$$\text{Value function:} \quad \hat{q}_t = Q(\mathbf{z}_t, \mathbf{a}_t) \tag{12}$$

$$\text{Policy prior:} \quad \hat{\mathbf{a}}_t = \pi(\mathbf{z}_t) \tag{13}$$

$$\text{CDRED model:} \quad \hat{r}_t = R(\mathbf{z}_t, \mathbf{a}_t) \tag{14}$$

The reward model, i.e., the CDRED model, consists of two predictors and $K$ target networks, estimating the expert and behavioral distributions for reward approximation. The encoder $h : \mathcal{S} \to \mathcal{Z}$ maps the observation (state-based or vision based) to latent representation. The latent dynamics model $d : \mathcal{Z} \times \mathcal{A} \to \mathcal{Z}$ learns the transition dynamics over the latent representations, implicitly modeling the environment dynamics. The value function learns to estimate the future return by training on temporal difference objective with the assist of the estimated rewards from the CDRED model. The policy prior learns a stochastic policy which guides the planning process of the world model. The training procedure is outlined in Algorithm 1, while the planning process is detailed in Algorithm 2.

**Model Training** The learnable parameters of the world model are denoted as three parts. While $\phi$ and $\psi$ denote the parameterization of expert predictor and behavioral predictor in the CDRED reward model, the rest of the parameters related to the encoder, latent dynamics, value model and policy prior are represented as $\xi$. Note that the parameters of the target networks $\bar{\theta}_k$ are not learnable. We train the encoder, dynamics model, value model, and reward model jointly with the following objective:

$$\mathcal{L}(\phi, \psi, \xi) = \sum_{t=0}^{H} \mathbb{E}_{(\mathbf{s}_t, \mathbf{a}_t, \mathbf{s}'_t) \sim \mathcal{B}_E \cup \mathcal{B}_\pi} \underbrace{\left[ \lambda^t \Big( \|\mathbf{z}'_t - \text{sg}(h(\mathbf{s}'_t))\|_2^2 + \text{CE}(\hat{q}_t, q_t) \Big) \right]}_{\text{Consistency and TD Loss}} + \underbrace{\mathcal{L}^r(\phi, \psi)}_{\text{CDRED Loss}} \tag{15}$$

The first term contains consistency loss and temporal difference loss to ensure the prediction consistency of the dynamics model and the accuracy for value function estimation. the temporal difference target is computed by $q_t = R(\mathbf{z}_t, \mathbf{a}_t) + \gamma Q(\mathbf{z}'_t, \pi(\mathbf{z}'_t))$ where $R(\mathbf{z}_t, \mathbf{a}_t)$ is the output of the reward model. We convert the regression TD objective into a classification problem for stabler value estimation, which is also used by the TD-MPC series and mentioned by Farebrother et al. (2024). $\mathrm{CE}(\hat{q}_t, q_t)$ is the cross entropy loss between target Q value and current predicted value. The second term is the reward loss, which is shown in Eq.6. Similar to the computation of reward loss, we also sum up the consistency and TD loss with factor $\lambda$ over a horizon $H$.

**Policy Prior Learning**   Regarding the policy prior update, we adopt maximum entropy objective (Haarnoja et al., 2018) to train a stochastic policy:

$$\mathcal{L}^\pi(\xi) = \sum_{t=0}^{H} \lambda^t \left[ \mathbb{E}_{(\mathbf{s}_t, \mathbf{a}_t) \sim \mathcal{B}_E \cup \mathcal{B}_\pi} \left[ -Q(\mathbf{z}_t, \pi(\mathbf{z}_t)) + \beta \log(\pi(\cdot|\mathbf{z}_t)) \right] \right] \tag{16}$$

We use short trajectories with horizon $H$ sampled from both expert and behavioral buffers for policy updates. We sum up the policy loss over the horizon with the same discount factor $\lambda$. $\beta$ is a fixed scalar coefficient to balance the entropy term and the Q value.

**Planning**   Following TD-MPC series (Hansen et al., 2022; 2023), we also leverage model predictive path integral (MPPI) (Williams et al., 2015) for planning. We optimize using the sampled action sequences $(\mathbf{a}_t, \mathbf{a}_{t+1}, ..., \mathbf{a}_{t+H})$ in a derivative-free style, maximizing the estimated return for the latent trajectories that have been rolled out using our dynamics model. Mathematically, our objective can be describe as a return maximizing process (Hansen et al., 2023):

$$\mu^*, \sigma^* = \underset{(\mu, \sigma)}{\mathrm{argmax}} \; \mathbb{E}_{(\mathbf{a}_t, \mathbf{a}_{t+1}, ..., \mathbf{a}_{t+H}) \sim \mathcal{N}(\mu, \sigma^2)} \left[ \gamma^H Q(\mathbf{z}_{t+H}, \mathbf{a}_{t+H}) + \sum_{h=t}^{H-1} \gamma^h R(\mathbf{z}_h, \mathbf{a}_h) \right] \tag{17}$$

After planning, the agent interacts with the environment using the first action $\mathbf{a}_t \sim \mathcal{N}(\mu^*, (\sigma^*)^2)$ to obtain new observations. New trajectories are stored in behavioral buffer $\mathcal{B}_\pi$ for following training.

# 4 EXPERIMENTS

We conduct experiments across a diverse range of tasks, including locomotion, manipulation, and tasks with both visual and state-based observations. We evaluate our approach using the DMControl (Tassa et al., 2018), Meta-World (Yu et al., 2020a) and ManiSkill2 (Gu et al., 2023) environments. As for the baselines, we compare our approach with IQ-MPC (Li et al., 2024), which integrates a world model architecture, as well as with model-free approaches, specifically IQ-Learn+SAC (Garg et al., 2021) (referred to as IQL+SAC in the plots), CFIL+SAC (Freund et al., 2023), HyPE (Ren et al., 2024) (In Appendix E.4), SAIL (Wang et al., 2020) (In Appendix E.5), and a fully offline baseline behavioral cloning (BC). Beyond standard comparisons with existing baselines, we further conduct additional experiments to provide a deeper understanding of our approach along the following aspects: (i): We conduct ablation studies on the number of expert trajectories, the choice of function $g$, the usage of world models and the usage of model predictive control, as detailed in Appendix E.2. (ii): We evaluate the exploration ability of our proposed approach (Appendix E.3), the robustness of our algorithm in noisy environment dynamics (Appendix E.6), examine the benefits of constructing the reward model in the latent space (Appendix E.9), and highlight its advantages over existing adversarial training methods (Appendix E.8). (iii): We provide the quantitative results measuring the training stability in Appendix E.7. For all experiments, we sample expert trajectories from a trained TD-MPC2 (Hansen et al., 2023). All of the experiments are conducted on a single RTX3090 graphic card.

## 4.1 META-WORLD EXPERIMENTS

We conduct experiments on 6 tasks in Meta-World environments. We use 100 expert trajectories for each task, ensuring that the expert data remains consistent across all algorithms for fair comparison within each task. IQ-MPC suffers from overly powerful discriminators in these tasks, even with gradient penalty applied, due to the adversarial training methodology. CFIL+SAC (Freund et al.,

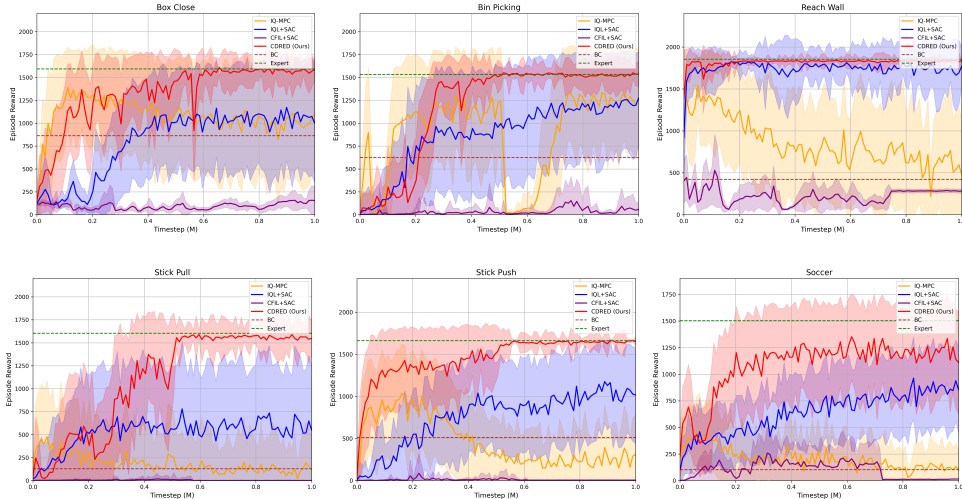

Figure 4: **Meta-World Results** We evaluate our CDRED method (red lines) on 6 tasks in Meta-World environments. We show stabler performance on these tasks, outperforming the baselines. IQ-MPC (orange lines) suffers from overly powerful discriminator problem mentioned in Section E.8. We conduct the experiments on 3 random seeds.

2023) encounters instability in the training process due to the challenges inherent in training flow models. We show stable and expert-level performance, outperforming these baselines in these tasks. We show the episode reward results in Figure 4 and success rate results in Table 1.

| Method | BC | IQL+SAC | CFIL+SAC | IQ-MPC | CDRED(Ours) |
|---|---|---|---|---|---|
| Box Close | 0.58±0.12 | 0.61±0.09 | 0.00±0.00 | 0.53±0.18 | **0.96±0.03** |
| Bin Picking | 0.43±0.18 | 0.75 ± 0.11 | 0.01±0.01 | 0.79±0.05 | **0.99±0.01** |
| Reach Wall | 0.10±0.08 | 0.90±0.04 | 0.01±0.01 | 0.31±0.14 | **0.98±0.01** |
| Stick Pull | 0.02±0.02 | 0.34±0.11 | 0.00±0.00 | 0.13±0.08 | **0.92±0.05** |
| Stick Push | 0.42±0.14 | 0.76±0.14 | 0.00±0.00 | 0.23±0.10 | **0.94±0.03** |
| Soccer | 0.04±0.03 | 0.73±0.09 | 0.01±0.01 | 0.12±0.07 | **0.81±0.05** |

Table 1: **Manipulation Success Rate Results in Meta-World** We show the success rate comparison on 6 tasks in Meta-World. Our CDRED model demonstrates outperforming results compared to existing methods. We compute the success rates over 100 episodes. We evaluate our model and other baselines on 3 random seeds.

## 4.2 DMCONTROL EXPERIMENTS

We conduct experiments on 6 tasks in DMControl (Tassa et al., 2018) environments. For low-dimensional tasks, we utilize 100 expert trajectories, while for high-dimensional tasks, we use 500 expert trajectories. Details on environment dimensionality can be found in Appendix D. Our CDRED model performs comparably to IQ-MPC on the Hopper Hop, Walker Run, and Humanoid Walk tasks. However, in Cheetah Run and Dog Stand, IQ-MPC experiences long-term instability, causing the agent to fail after extensive online training. On the Reacher Hard task, IQ-MPC struggles with an overly powerful discriminator, which prevents it from learning an expert-level policy. The model-free methods in baseline algorithms fail to achieve stable, expert-level performance on these tasks. The episode reward results are shown in Figure 5.

## 4.3 VISION-BASED EXPERIMENTS

In addition to experiments using state-based observations, we also benchmark our method on tasks with visual observations. Specifically, we select three tasks from DMControl (Tassa et al., 2018)

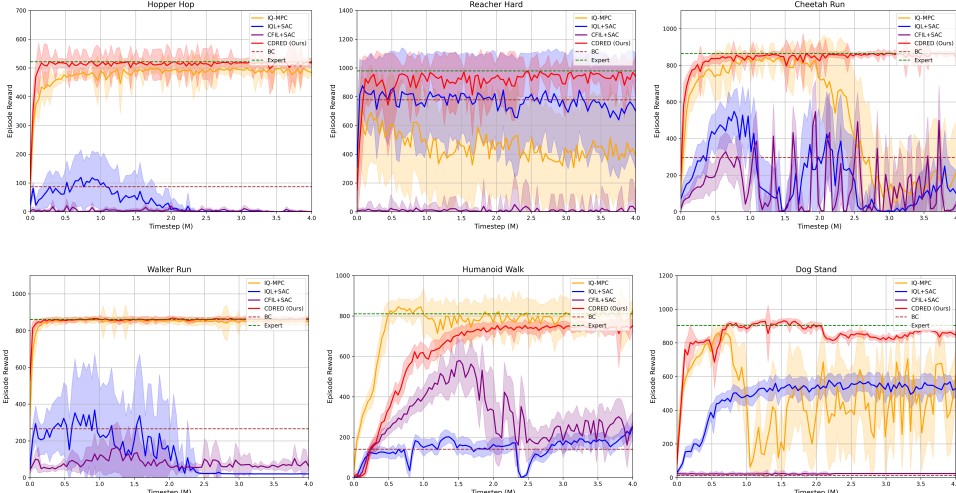

Figure 5: **DMControl Results** We evaluate our CDRED method (red lines) on 6 tasks in DMControl environments. Our approach achieves results comparable to IQ-MPC (orange lines) in Hopper Hop, Walker Run, and Humanoid Walk, while demonstrating greater stability across the remaining tasks. We conduct the experiments on 3 random seeds.

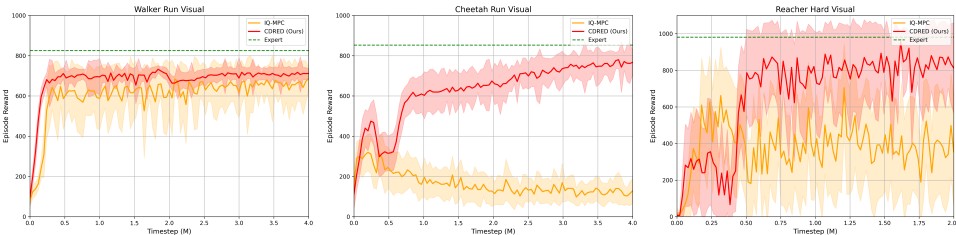

Figure 6: **Visual Experiment Results** We compare the results of our model with IQ-MPC on tasks with visual observations. Our approach outperforms IQ-MPC in Cheetah Run and Reacher Hard tasks, while obtains comparable performance on Walker Run task. We conduct the experiments on 3 random seeds.

with visual observations. To create these visual datasets, we render visual observations based on state-based expert trajectories, replacing the original state-based observations in the expert data. For each task, we use 100 expert trajectories generated by a trained TD-MPC2 model (Hansen et al., 2023). We show our results in Figure 6. Interestingly, we observe that visual IQ-MPC encounters an issue with an overly powerful discriminator in the Cheetah Run task when using trajectories generated by a trained state-based TD-MPC2 policy, where state observations are replaced by RGB images rendered from those states. However, IQ-MPC performs well when using expert trajectories generated by a TD-MPC2 policy trained directly on visual observations.

## 5 CONCLUSION

We propose a novel approach for world model-based online imitation learning, featuring an innovative reward model formulation. Unlike traditional adversarial approaches that may introduce instability during training, our reward model is grounded in density estimation for both expert and behavioral state-action distributions. This formulation enhances stability while maintaining high performance. Our model demonstrates expert-level proficiency across various tasks in multiple benchmarks, including DMControl, Meta-World, and ManiSkill2. Furthermore, it consistently retains stable performance throughout long-term online training. With its robust reward modeling and stability, our approach has the potential to tackle complex real-world robotics control tasks, where reliability and adaptability are crucial.

## REPRODUCIBILITY STATEMENT

We have made significant efforts to ensure the reproducibility of our results. The hyperparameter settings and architectural details are documented in Appendix B, while the training and planning algorithms are described in Appendix C. Finally, our source code is provided to facilitate faithful reproduction of our experiments in the supplementary materials.

## ETHICS STATEMENT

We have carefully reviewed the Code of Ethics and find that our work does not raise any significant ethical concerns. Our research does not involve human subjects, sensitive data, or potentially harmful applications. We believe our methodology and contributions align with principles of fairness, transparency, and research integrity.

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

## A    RELATED WORKS

Our work builds on previous advancements in imitation learning and model-based reinforcement learning.

**Imitation Learning**    Recent advancements in imitation learning (IL) have leveraged deep neural networks and diverse methodologies to enhance performance. Generative Adversarial Imitation Learning (GAIL) (Ho & Ermon, 2016) laid the foundation for adversarial reward learning by formulating it as a min-max optimization problem inspired by Generative Adversarial Networks (GANs) (Goodfellow et al., 2014). Several approaches have built on GAIL. Model-based Adversarial Imitation Learning (MAIL) (Baram et al., 2016) extended GAIL with a forward model trained via data-driven methods. ValueDICE (Kostrikov et al., 2019) transformed the adversarial framework by focusing on off-policy learning through distribution ratio estimation.

Offline imitation learning has seen significant advancements through approaches like Diffusion Policy (Chi et al., 2023), which applied diffusion models for behavioral cloning, and Ditto (DeMoss et al., 2023), which combined Dreamer V2 (Hafner et al., 2020) with adversarial techniques. Implicit BC (Florence et al., 2022) demonstrated that supervised policy learning with implicit models improves empirical performance in robotic tasks. DMIL (Zhang et al., 2023) leveraged a discriminator to assess dynamics accuracy and the suboptimality of model rollouts against expert demonstrations in offline IL.

Other innovations focused on integrating advanced reinforcement learning techniques. Inverse Soft Q-Learning (IQ-Learn) (Garg et al., 2021) reformulated GAIL's learning objectives, applying them to soft actor-critic (Haarnoja et al., 2018) and soft Q-learning agents. SQIL (Reddy et al., 2019) contributed an online imitation learning algorithm utilizing soft Q-functions. CFIL (Freund et al., 2023) introduced a coupled flow method for simultaneous reward generation and policy learning from expert demonstrations. Random Expert Distillation (RED) (Wang et al., 2019) proposed an alternative method for constructing reward models by estimating the support of the expert policy distribution.

Model-based methods have also played a pivotal role in advancing IL. V-MAIL (Rafailov et al., 2021) employed variational models to facilitate imitation learning, while CMIL (Kolev et al., 2024) utilized conservative world models for image-based manipulation tasks. Prior works (Englert et al., 2013; Hu et al., 2022; Igl et al., 2022) highlighted the potential of model-based imitation learning in real-world robotics control and autonomous driving. A model-based inverse reinforcement learning approach by Das et al. (2021) explored key-point prediction to improve performance in imitation tasks. Hybrid Inverse Reinforcement Learning (Ren et al., 2024) offered a novel strategy blending online and expert demonstrations, enhancing agent robustness in stochastic settings. EfficientImitate (Yin et al., 2022) fused EfficientZero (Ye et al., 2021) with adversarial imitation learning, achieving impressive performance on DMControl tasks (Tassa et al., 2018).

**Model-based Reinforcement Learning**    Recent advancements in model-based reinforcement learning (MBRL) utilize learned dynamics models, constructed via data-driven methodologies, to enhance agent learning and decision-making. MBPO (Janner et al., 2019) introduced a model-based policy optimization algorithm that ensures stepwise monotonic improvement. Extending this to offline RL, MOPO (Yu et al., 2020b) incorporated a penalty term in the reward function based on the uncertainty of the dynamics model to manage distributional shifts effectively. MBVE (Feinberg et al., 2018) augmented model-free agents with model-based rollouts to improve value estimation.

Many approaches focus on constructing dynamics models in latent spaces. PlaNet (Hafner et al., 2019b) pioneered this direction by proposing a recurrent state-space model (RSSM) with an evidence lower bound (ELBO) training objective, addressing challenges in partially observed Markov decision processes (POMDPs). Building on PlaNet, the Dreamer algorithms (Hafner et al., 2019a; 2020; 2023) leveraged learned world models to simulate future trajectories in a latent space, enabling efficient learning and planning. The TD-MPC series (Hansen et al., 2022; 2023) further refined latent-space modeling by developing a scalable world model for model predictive control, utilizing a temporal-difference learning objective to improve performance. Similarly, MuZero (Schrittwieser et al., 2020) combined a latent dynamics model with tree-based search to achieve strong performance in discrete control tasks, blending planning and policy learning seamlessly. The EfficientZero series

(Ye et al., 2021; Wang et al., 2024) enhances MuZero, achieving superior sampling efficiency in visual reinforcement learning tasks.

## B HYPERPARAMETERS AND ARCHITECTURAL DETAILS

### B.1 ARCHITECTURAL DETAILS

We show the overall model architecture via a Pytorch style notation. We leverage layernorm (Ba, 2016) and Mish activations (Misra, 2019) for our model. The detialed architecture is displayed as following:

```
WorldModel(
    (_encoder): ModuleDict(
        (state): Sequential(
            (0): NormedLinear(in_features=state_dim, out_features=256,
                bias=True, act=Mish)
            (1): NormedLinear(in_features=256, out_features=512, bias=
                True, act=SimNorm)
        )
    )
    (_dynamics): Sequential(
        (0): NormedLinear(in_features=512+action_dim, out_features=512,
            bias=True, act=Mish)
        (1): NormedLinear(in_features=512, out_features=512, bias=True,
            act=Mish)
        (2): NormedLinear(in_features=512, out_features=512, bias=True,
            act=SimNorm)
    )
    (_reward): CDRED_Reward(
        (behavioral_predictor): Sequential(
            (0): NormedLinear(in_features=512+action_dim, out_features
                =512, bias=True, act=Mish)
            (1): NormedLinear(in_features=512, out_features=512, bias=
                True, act=Mish)
            (2): Linear(in_features=512, out_features=64, bias=True)
        )
        (expert_predictor): Sequential(
            (0): NormedLinear(in_features=512+action_dim, out_features
                =512, bias=True, act=Mish)
            (1): NormedLinear(in_features=512, out_features=512, bias=
                True, act=Mish)
            (2): Linear(in_features=512, out_features=64, bias=True)
        )
        (target_networks)[not learnable]: Vectorized ModuleList(
            (0-4): 5 x Sequential(
                (0): NormedLinear(in_features=512+action_dim,
                    out_features=512, bias=True, act=Mish)
                (1): NormedLinear(in_features=512, out_features=512, bias
                    =True, act=Mish)
                (2): Linear(in_features=512, out_features=64, bias=True)
            )
        )
    )
    (_pi): Sequential(
        (0): NormedLinear(in_features=512, out_features=512, bias=True,
            act=Mish)
        (1): NormedLinear(in_features=512, out_features=512, bias=True,
            act=Mish)
        (2): Linear(in_features=512, out_features=2*action_dim, bias=True
            )
    )
    (_Qs): Vectorized ModuleList(
        (0-4): 5 x Sequential(
```

```
            (0): NormedLinear(in_features=512+action_dim, out_features
                =512, bias=True, dropout=0.01, act=Mish)
            (1): NormedLinear(in_features=512, out_features=512, bias=
                True, act=Mish)
            (2): Linear(in_features=512, out_features=101, bias=True)
        )
    )
    (_target_Qs): Vectorized ModuleList(
        (0-4): 5 x Sequential(
            (0): NormedLinear(in_features=512+action_dim, out_features
                =512, bias=True, dropout=0.01, act=Mish)
            (1): NormedLinear(in_features=512, out_features=512, bias=
                True, act=Mish)
            (2): Linear(in_features=512, out_features=num_bins, bias=True
                )
        )
    )
)
```

## B.2 HYPERPARAMETER DETAILS

The specific hyperparameters used in the CDRED reward model are as follows:

- The predictors and target networks project latent state-action pairs to an embedding space with dimension $p = 64$.
- We use an ensemble of 5 target networks for the CDRED reward model.
- The function $g(x) = x$ is used in all experiments.
- The value of $\zeta = 0.8$ is used across all experiments.
- We adopt $\alpha = 0.9$ for all experiments.
- A StepLR learning rate scheduler is employed with $\gamma_{\text{lr}} = 0.1$, with a scheduler step of $500K$ for Meta-World and ManiSkill2 experiments, and $2M$ for DMControl experiments.

The remaining hyperparameters are consistent with those used in TD-MPC2 (Hansen et al., 2023).

# C  TRAINING AND PLANNING ALGORITHMS

## C.1  TRAINING ALGORITHM

In this section, we present the detailed training algorithm for the CDRED world model, as shown in Algorithm 1. For clarity, let $\theta = \{\phi, \psi, \xi\}$ represent all learnable parameters of the world model, and $\theta^-$ denote a fixed copy of $\theta$.

---

**Algorithm 1**  CDRED World Model (*training*)

---

**Require:** $\theta, \theta^-$: randomly initialized network parameters

      $\eta, \tau, \lambda, \mathcal{B}_\pi, \mathcal{B}_E$: learning rate, soft update coefficient, horizon discount coefficient, behavioral buffer, expert buffer

  **for** training steps **do**

    *// Collect episode with CDRED world model from $\mathbf{s}_0 \sim p_0$:*

    **for** step $t = 0...T$ **do**

      Compute $\mathbf{a}_t$ with $\pi_\theta(\cdot | h_\theta(\mathbf{s}_t))$ using Algorithm 2         ◁ *Planning with MPPI*

      $(\mathbf{s}_t', r_t) \sim$ env.step($\mathbf{a}_t$)

      $\mathcal{B}_\pi \leftarrow \mathcal{B}_\pi \cup (\mathbf{s}_t, \mathbf{a}_t, r_t, \mathbf{s}_t')$         ◁ *Add to behavioral buffer*

      $\mathbf{s}_{t+1} \leftarrow \mathbf{s}_t'$

    **end for**

    *// Update reward-free world model using collected data in $\mathcal{B}_\pi$ and $\mathcal{B}_E$:*

    **for** num updates per step **do**

      $(\mathbf{s}_t, \mathbf{a}_t, \mathbf{s}_t')_{0:H} \sim \mathcal{B}_\pi \cup \mathcal{B}_E$         ◁ *Combine behavioral and expert batch*

      $\mathbf{z}_0 = h_\theta(\mathbf{s}_0)$         ◁ *Encode first observation*

      *// Unroll for horizon H*

      **for** $t = 0...H$ **do**

        $\mathbf{z}_{t+1} = d_\theta(\mathbf{z}_t, \mathbf{a}_t)$         ◁ *Unrolling using the latent dynamics model*

        $\hat{q}_t = Q(\mathbf{z}_t, \mathbf{a}_t)$         ◁ *Estimate the Q value*

        $\mathbf{z}_t' = h(\mathbf{s}_t')$         ◁ *Encode the ground-truth next state*

        $\hat{r}_t = R(\mathbf{z}_t, \mathbf{a}_t)$         ◁ *Estimate Reward using the CDRED reward model*

        $q_t = \hat{r}_t + \gamma Q(\mathbf{z}_t', \pi(\mathbf{z}_t'))$         ◁ *Compute the TD target using the estimated reward*

      **end for**

      Compute model loss $\mathcal{L}$         ◁ *Equation 15*

      Compute policy prior loss $\mathcal{L}^\pi$         ◁ *Equation 16*

      $\theta \leftarrow \theta - \frac{1}{H}\eta\nabla_\theta(\mathcal{L} + \mathcal{L}^\pi)$         ◁ *Update online network*

      $\theta^- \leftarrow (1 - \tau)\theta^- + \tau\theta$         ◁ *Soft update*

    **end for**

  **end for**

---

## C.2 PLANNING ALGORITHM

In this section, we present the detailed MPPI planning algorithm for the CDRED world model, as shown in Algorithm 2. For simplicity, let $\theta = \{\phi, \psi, \xi\}$ represent all learnable parameters of the world model.

---

**Algorithm 2** CDRED World Model (*inference*)

---

**Require:** $\theta$ : learned network parameters
        $\mu^0, \sigma^0$: initial parameters for $\mathcal{N}$
        $N, N_\pi$: number of sample/policy trajectories
        $\mathbf{s}_t, H$: current state, rollout horizon
1: Encode state $\mathbf{z}_t \leftarrow h_\theta(\mathbf{s}_t)$
2: **for** each iteration $j = 1..J$ **do**
3:     Sample $N$ trajectories of length $H$ from $\mathcal{N}(\mu^{j-1}, (\sigma^{j-1})^2 \mathrm{I})$
4:     Sample $N_\pi$ trajectories of length $H$ using $\pi_\theta, d_\theta$
      *// Estimate trajectory returns $\phi_\Gamma$ using $d_\theta, Q_\theta, \pi_\theta, R_\theta$ starting from $\mathbf{z}_t$ and initialize $\phi_\Gamma = 0$:*

5:     **for** all $N + N_\pi$ trajectories $(\mathbf{a}_t, \mathbf{a}_{t+1}, \ldots, \mathbf{a}_{t+H})$ **do**
6:       **for** step $t = 0..H - 1$ **do**
7:         $\mathbf{z}_{t+1} \leftarrow d_\theta(\mathbf{z}_t, \mathbf{a}_t)$                $\triangleleft$ *Latent transition*
8:         $\hat{\mathbf{a}}_{t+1} \sim \pi_\theta(\cdot | \mathbf{z}_{t+1})$
9:         $\phi_\Gamma = \phi_\Gamma + \gamma^t R_\theta(\mathbf{z}_t, \mathbf{a}_t)$      $\triangleleft$ *Estimate reward with CDRED reward model*
10:       **end for**
11:       $\phi_\Gamma = \phi_\Gamma + \gamma^H Q_\theta(\mathbf{z}_H, \mathbf{a}_H)$           $\triangleleft$ *Terminal Q value*
12:     **end for**
13:     *// Update parameters $\mu, \sigma$ for next iteration:*
14:     $\mu^j, \sigma^j \leftarrow$ MPPI update with $\phi_\Gamma$.
15: **end for**
16: **return** $\mathbf{a} \sim \mathcal{N}(\mu^J, (\sigma^J)^2 \mathrm{I})$

---

# D  TASK DETAILS AND ENVIRONMENT SPECIFICATIONS

We consider 12 continuous control tasks in locomotion control and robot manipulation. We leverage 6 manipulation tasks in Meta-World (Yu et al., 2020a), 6 locomotion tasks in DMControl (Tassa et al., 2018) and 3 tasks in ManiSKill2 (Gu et al., 2023). In this section, we list the environment specifications for completeness in Table 2, Table 3 and Table 4.

| Task | Observation Dimension | Action Dimension |
|:---:|:---:|:---:|
| Box Close | 39 | 4 |
| Bin Picking | 39 | 4 |
| Reach Wall | 39 | 4 |
| Stick Pull | 39 | 4 |
| Stick Push | 39 | 4 |
| Soccer | 39 | 4 |

Table 2: **Meta-World Tasks** We evaluate on 6 tasks in Meta-World. The Meta-World benchmark is specifically constructed to facilitate research in multitask and meta-learning, ensuring a consistent embodiment, observation space, and action space across all tasks.

| Task | Observation Dimension | Action Dimension | High-dimensional? |
|---|---|---|---|
| Reacher Hard | 6 | 2 | No |
| Hopper Hop | 15 | 4 | No |
| Cheetah Run | 17 | 6 | No |
| Walker Run | 24 | 6 | No |
| Humanoid Walk | 67 | 24 | Yes |
| Dog Stand | 223 | 38 | Yes |

Table 3: **DMControl Tasks** We evaluate on 6 tasks in DMControl. DMControl is a benchmark for reinforcement learning, offering a range of continuous control tasks built on the MuJoCo physics engine. It provides diverse environments for testing algorithms on tasks from basic motions to complex behaviors, supporting standardized evaluation in control and planning research.

| Task | Observation Dimension | Action Dimension |
|---|---|---|
| Lift Cube | 42 | 4 |
| Pick Cube | 51 | 4 |
| Turn Faucet | 40 | 7 |

Table 4: **ManiSkill2 Tasks** We evaluate on 3 tasks in ManiSkill2. The ManiSkill2 benchmark represents a sophisticated platform designed to advance large-scale robot learning capabilities. It distinguishes itself through comprehensive task randomization and an extensive array of task variations, enabling more robust and generalized robotic skill development.

# E  ADDITIONAL EXPERIMENTS

## E.1  EXPERIMENTS ON MANISKILL2

We further evaluate our method on additional manipulation tasks in ManiSkill2 (Gu et al., 2023), achieving stable and competitive results on the pick cube, lift cube, and turn faucet tasks. Notably, IQL+SAC (Garg et al., 2021) and IQ-MPC (Li et al., 2024) also perform relatively well in these scenarios. Table 5 summarizes the success rates of each method across the ManiSkill2 tasks.

| Method | IQL+SAC | CFIL+SAC | IQ-MPC | CDRED(Ours) |
|---|---|---|---|---|
| Pick Cube | 0.61±0.13 | 0.00±0.00 | 0.79±0.05 | **0.87±0.04** |
| Lift Cube | 0.85 ± 0.04 | 0.01±0.01 | 0.89±0.02 | **0.93±0.03** |
| Turn Faucet | 0.82±0.04 | 0.00±0.00 | 0.73±0.08 | **0.84±0.08** |

Table 5: **Manipulation Success Rate Results in ManiSkill2** We evaluate the success rate of CDRED across three tasks in the ManiSkill2 environment. CDRED demonstrates superior performance compared to IQL+SAC, CFIL+SAC, and IQ-MPC on the Pick Cube and Lift Cube tasks, while achieving comparable results on Turn Faucet. The reported results are averaged over 100 trajectories and evaluated across three random seeds.

## E.2  ABLATION STUDIES

To evaluate the influence of different architecture choices and expert data amounts, we ablate over the expert trajectories number, the $g$ function choice, and the usage of coupling. We show that our approach is still robust under a small number of expert demonstrations.

**Ablation on Expert Trajectories Number**  We evaluate the impact of the number of expert trajectories on model performance and find that our model can learn effectively with a limited number of expert trajectories. We conduct this ablation on the Bin Picking task in Meta-World and the Cheetah

Run task in DMControl, observing that our model achieves expert-level performance with only five demonstrations. The results are presented in Figure 7. Our model can effectively learn with only 5 expert demonstrations for Cheetah Run and Bin Picking tasks.

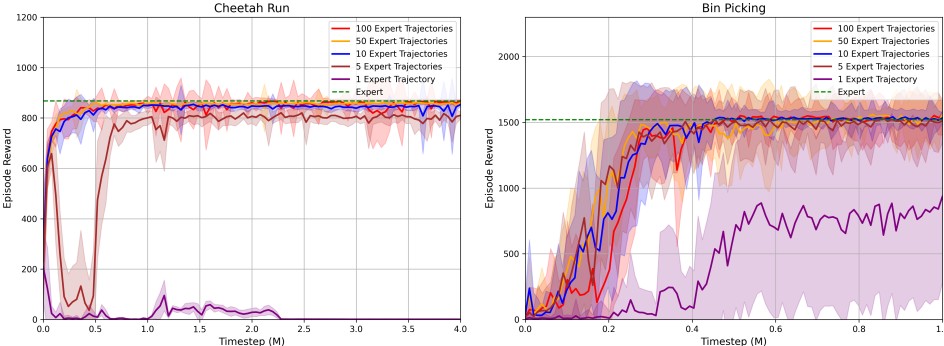

Figure 7: **Ablation Study on Expert Trajectories Number** We conduct an ablation study on the number of expert trajectories for the Cheetah Run task in DMControl and the Bin Picking task in Meta-World. Our results demonstrate that our model can achieve expert-level performance using only 5 expert demonstrations for both tasks.

**Ablation on the $g$ Function Choice** Function $g$ maps the neural network output bonus to the actual reward space. In order to keep the optimal point for the reward function unchanged, we need to leverage a monotonically increasing function. Empirically, we find $g(x) = x$ and $g(x) = \exp(x)$ can both work, but they have different performances in high-dimensional settings. We find $g(x) = x$ tends to provide a faster convergence in high-dimensional tasks such as Dog Stand compared to $g(x) = \exp(x)$. While we haven't observed any significant difference on low-dimensional tasks. We show the ablation in Figure 8.

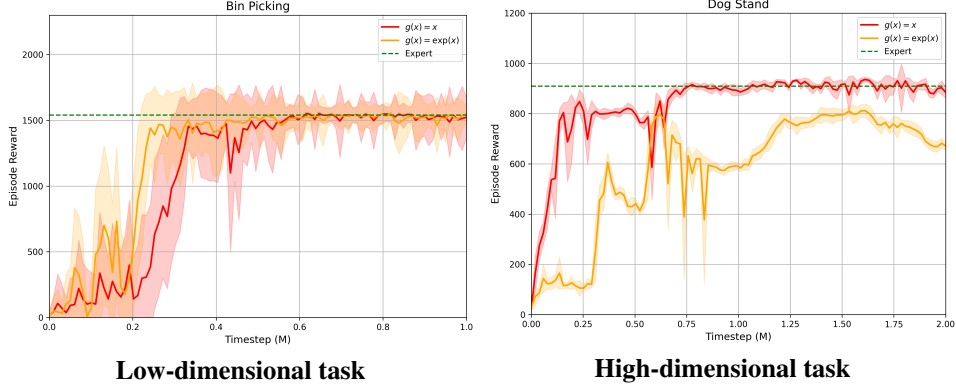

**Low-dimensional task**                    **High-dimensional task**

Figure 8: **Ablation on $g$ function choice** For low-dimensional task (left), both forms of $g(x)$ demonstrate comparable performance. However, in high-dimensional task (right), $g(x) = \exp(x)$ exhibits instability and suboptimal behavior, whereas $g(x) = x$ maintains stability. The task dimensionality information is shown in Appendix D.

**Ablation on the Hyperparameter Choice** We conduct ablation studies on two hyperparameters, $\alpha$ and $\zeta$, introduced in Section 3.2, which are related to the construction of the reward model. Our experiments demonstrate that these parameters influence the model's convergence during the initial training phase, which is closely tied to the policy's exploration capability. For the hyperparameter $\zeta$, we find that smaller values may encourage exploration, leading to faster convergence. However, if $\zeta$ is too small, the model may fail to learn effectively. For the hyperparameter $\alpha$, larger values may enhance exploration, potentially promoting convergence. The results are aligned with our intuition given in Section 3.2. We perform the ablation study on the state-based Humanoid Walk task in the DMControl environment, and the results are presented in Figure 9.

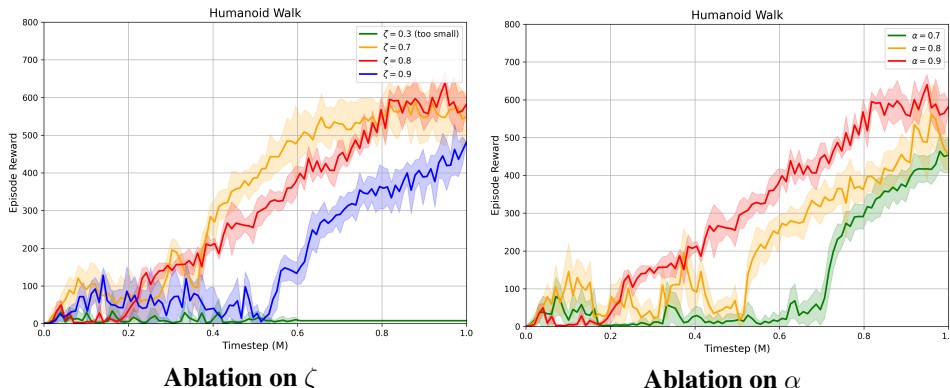

**Ablation on $\zeta$**                    **Ablation on $\alpha$**

Figure 9: **Ablation Study on Hyperparameters** We conduct an ablation study on hyperparameter $\zeta$ and $\alpha$. The ablation study is conducted on Humanoid Walk task.

**Ablation on World Models** To assess the impact of model-based learning, we conduct an ablation study comparing performance with and without the world model. In the ablated setting, we train SAC (Haarnoja et al., 2018) directly in the latent space using only the CDRED reward model, without leveraging the world model. The results, including sampling complexity, episode rewards, and success rate are presented in Table 6.

| | Sampling Complexity | | Success rate / Reward | |
|---|---|---|---|---|
| **Task** | **w/ world model** | **w/o world model** | **w/ world model** | **w/o world model** |
| Walker Run | $\sim$**150k** | $\sim$1.2M | **856.3 $\pm$ 5.5** | 741.9 $\pm$ 14.8 |
| Bin Picking | $\sim$**500k** | $\sim$1M | **0.99 $\pm$ 0.01** | 0.83 $\pm$ 0.06 |

Table 6: **Ablation on World Models** Our ablation study demonstrates that using a world model significantly improves performance and reduces sampling complexity. We evaluated success rate (Bin Picking) and episode rewards (Walker Run), averaging all results across 3 random seeds.

**Ablation on Model Predictive Control** Our ablation study of the Model Predictive Control (MPC) component (Algorithm 2), summarized in Table 7, reveals consistent performance improvements across all tasks, with the most significant gains in high-dimensional environments like Dog Stand.

| Task | w/ MPC | w/o MPC |
|---|---|---|
| Bin Picking | $0.99 \pm 0.01$ | $0.95 \pm 0.02$ |
| Stick Push | $0.94 \pm 0.03$ | $0.91 \pm 0.05$ |
| Walker Run | $856.3 \pm 5.5$ | $837.1 \pm 4.8$ |
| Dog Stand | $915.6 \pm 12.3$ | $687.2 \pm 33.9$ |

Table 7: **Ablation Study on Model Predictive Control** Our ablation study on Model Predictive Control (MPC) reveals consistent performance improvements across all tasks. While all environments benefit, the high-dimensional Dog Stand task shows the most significant gains. These results, which measure success rates (Bin Picking, Stick Push) and episode rewards (Walker Run, Dog Stand), are averaged across 3 random seeds.

### E.3    ADDITIONAL EXPERIMENT ON EXPLORATION ABILITY

To verify the exploration improvements from using coupled estimators and to validate our initial toy experiment (Figure 1), we evaluated our method on the AntMaze tasks in the D4RL benchmark (Fu et al., 2020).

We performed an ablation study comparing performance with and without coupled estimators on the Umaze-diverse, Medium-diverse, and Large-diverse environments. The results, shown in Table 8, demonstrate that using a coupled estimator leads to significantly higher success rates, especially in the larger and more complex mazes that demand stronger exploration. This confirms that our approach enhances exploration capabilities.

| Task | w/ Coupling | w/o Coupling |
|---|---|---|
| Antmaze-Umaze-Diverse | $0.87 \pm 0.06$ | $0.82 \pm 0.04$ |
| Antmaze-Medium-Diverse | $0.67 \pm 0.04$ | $0.45 \pm 0.09$ |
| Antmaze-Large-Diverse | $0.52 \pm 0.08$ | $0.17 \pm 0.05$ |

Table 8: **Analysis on Exploration Ability with Coupled Estimator** We performed an ablation study on the AntMaze environments using 50 expert trajectories. The results, averaged across 3 random seeds, show that our coupled estimator yields the most significant improvements in larger mazes, where stronger exploration capabilities are critical.

### E.4 ADDITIONAL COMPARISON WITH HYPE

Hybrid IRL (Ren et al., 2024) is a recently proposed method for performing inverse reinforcement learning and imitation learning using hybrid data. In this section, we compare our approach with the model-free method (HyPE) introduced in their work. Our method achieves superior empirical performance on three DMControl locomotion tasks, including the high-dimensional Humanoid Walk task. The results are presented in Figure 10.

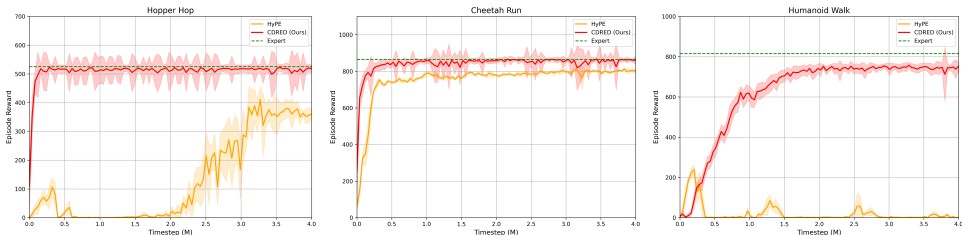

Figure 10: **Comparison with HyPE** We compare our CDRED approach with the HyPE method (Ren et al., 2024) on the Hopper Hop, Cheetah Run, and Humanoid Walk tasks. Among these, the Humanoid Walk task is high-dimensional, while the others are low-dimensional. Our approach demonstrates superior empirical performance and improved sampling efficiency on these tasks.

### E.5 ADDITIONAL COMPARISON WITH SAIL

Support-weighted Adversarial Imitation Learning (SAIL) (Wang et al., 2020) is an extension of Generative Adversarial Imitation Learning (GAIL) (Ho & Ermon, 2016) that enhances performance by integrating Random Expert Distillation (RED) rewards (Wang et al., 2019). In this section, we present an additional comparative analysis between our proposed CDRED method and SAIL. The experimental results are illustrated in Figure 11.

### E.6 ROBUSTNESS ANALYSIS UNDER NOISY DYNAMICS

We conduct an additional analysis to evaluate the robustness of our model under noisy environment dynamics. Following the evaluation protocol of Hybrid IRL (Ren et al., 2024), we introduce noise by adding a trembling probability, $p_{\text{tremble}}$. During interactions with the environment, the agent executes a random action with probability $p_{\text{tremble}}$ and follows the action generated by the policy for the remaining time. Our empirical results demonstrate that our model exhibits robustness to noisy dynamics, as its performance only slightly deteriorates from the expert level when noise is introduced. The results for the Cheetah Run and Walker Run tasks are presented in Figure 12.

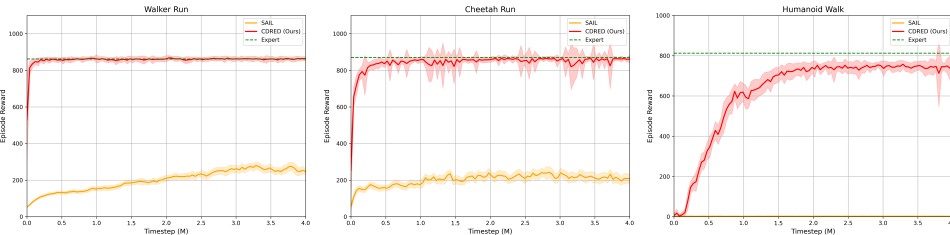

Figure 11: **Comparison with SAIL** We compare our CDRED approach with the SAIL method (Wang et al., 2020) on the Walker Run, Cheetah Run, and Humanoid Walk tasks. Among these, the Humanoid Walk task is high-dimensional, while the others are low-dimensional. SAIL fails to learn in the high-dimensional Humanoid Walk task while our approach achieves nearly expert-level performance. Overall, our approach demonstrates superior empirical performance and improved sampling efficiency on these tasks.

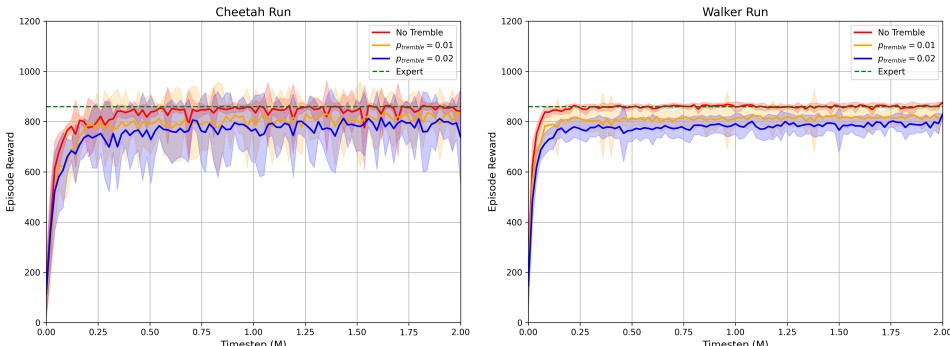

Figure 12: **Robustness Analysis under Noisy Environment Dynamics** We analyze the performance of our model on the Cheetah Run and Walker Run tasks under stochastic environment dynamics. Our results demonstrate that the model shows notable robustness to noise in the environment dynamics.

### E.7 QUANTITATIVE ANALYSIS OF TRAINING STABILITY

To assess the training stability of our algorithm, we examine the mean and maximum gradient norms throughout the training process. This approach is similar to the analysis conducted in TD-MPC2 (Hansen et al., 2023). We compare the gradient norms of our method with those of IQ-MPC (Li et al., 2024), a world model online imitation learning approach that employs an adversarial formulation, on DMControl tasks. Our results indicate that the gradient norms of our approach are significantly smaller than those of IQ-MPC, suggesting superior training stability. The detailed comparison is presented in Table 9.

| Gradient Norm | IQ-MPC (mean) | CDRED (mean) | IQ-MPC (max) | CDRED (max) |
|---|---|---|---|---|
| Humanoid Walk | 12.6 | 0.073 | 198.3 | 0.32 |
| Hopper Hop | 324.8 | 1.3 | 8538.6 | 4.6 |
| Cheetah Run | 131.7 | 0.34 | 2342.6 | 3.1 |
| Walker Run | 344.6 | 0.26 | 1534.7 | 1.8 |
| Reacher Hard | 11.3 | 0.012 | 65.8 | 0.083 |
| Dog Walk | 989.7 | 0.059 | 6824.3 | 0.13 |

Table 9: **Training Stability Analysis** Comparison of gradient norms between our CDRED approach and the IQ-MPC method. The significantly smaller gradient norms of our approach indicate enhanced training stability.

### E.8 ADVANTAGES COMPARED TO CURRENT METHODS INVOLVING ADVERSARIAL TRAINING

The current existing methods (Li et al., 2024; Kolev et al., 2024; Rafailov et al., 2021; Yin et al., 2022) for world model online imitation learning often involve adversarial training, following the similar problem formulation as GAIL (Ho & Ermon, 2016) or IQ-Learn (Garg et al., 2021). IQ-MPC (Li et al., 2024) adopted inverse soft-Q objective for critic learning while CMIL (Kolev et al., 2024), V-MAIL (Rafailov et al., 2021) and EfficientImitate (Yin et al., 2022) leveraged GAIL style reward modeling. In terms of IQ-Learn, an improved version of GAIL, although its policy can be computed by applying a softmax to the Q-value in discrete control, effectively converting a min-max problem into a single maximization (Garg et al., 2021), it still requires the maximum entropy RL objective for policy updates in continuous control settings. In such cases, IQ-Learn performs adversarial training between the policy and the critic, which leads to stability issues similar to those encountered in GAIL. IQ-MPC, while performing well in various complex scenarios such as high-dimensional locomotion control and dexterous hand manipulation, still encounters challenges in some cases. These challenges include an imbalance between the discriminator and the policy, as well as long-term instability. These issues stem from using an adversarially trained Q-function as the critic. While IQ-MPC attempts to mitigate them by incorporating regularization terms during the training process, it doesn't fully resolve the problem. Figure 13 illustrates the drawbacks of IQ-MPC in some cases, namely an overly powerful discriminator and long-term instability. We also demonstrate the quantitative results for training stability analysis in Appendix E.7.

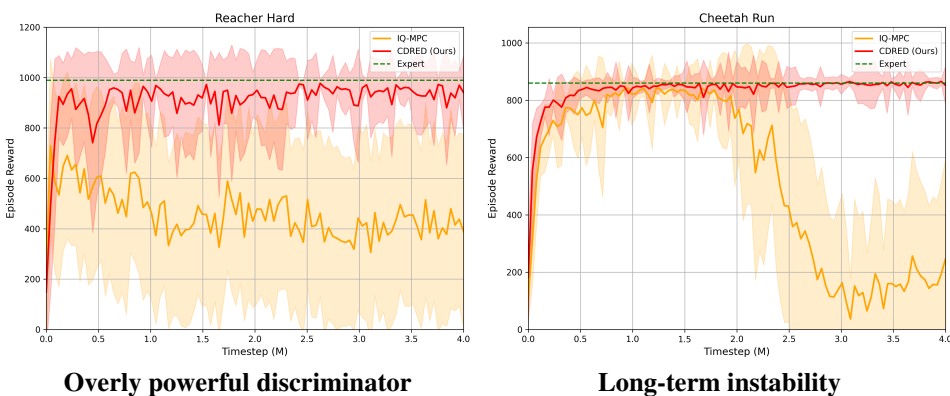

**Overly powerful discriminator**          **Long-term instability**

Figure 13: **Drawbacks of Methods Including Adversarial Training** We demonstrate the drawbacks of IQ-MPC (Li et al., 2024) in some tasks, which employs adversarial training for online imitation learning. An overly powerful discriminator (Left) leads to sub-optimal policy learning, while long-term instability (Right) of adversarial training prevents IQ-MPC from maintaining expert-level performance during extended online training. Our CDRED method, which replaces adversarial training with density estimation, is immune to these issues.

**Overly Powerful Discriminator**    The generative adversarial training process is often prone to instability (Gulrajani et al., 2017). IQ-MPC employs generative adversarial training between the policy and the critic, and it also encounters this challenge. To mitigate this issue, IQ-MPC leverages gradient penalty from Gulrajani et al. (2017) to enforce Lipschitz condition of the gradients in a form of:

$$\mathcal{L}^{pen} = \sum_{t=0}^{H} \lambda^t \left[ \mathbb{E}_{(\hat{\mathbf{s}}_t, \hat{\mathbf{a}}_t) \sim \mathcal{B}} \left( \|\nabla Q(\hat{\mathbf{z}}_t, \hat{\mathbf{a}}_t)\|_2 - 1 \right)^2 \right] \tag{18}$$

In the gradient penalty, $(\hat{\mathbf{s}}_t, \hat{\mathbf{a}}_t)$ are data points on straight lines between expert and behavioral distributions, which are generated by linear interpolation. Although it counters the problem to some extent, the performance of IQ-MPC is still not satisfactory in some tasks such as Reacher in DM-Control and Meta-World robotics manipulation tasks, for which we will refer to our experimental results in Section 4. An overly powerful discriminator often causes the Q-value difference between the policy and expert distributions to diverge, as noted by Li et al. (2024). Specifically, this divergence is reflected in the gap between the expected Q-values under the expert distribution,

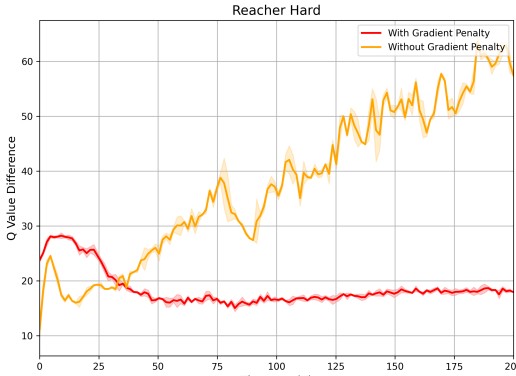

Figure 14: **IQ-MPC Q Value Difference Visualization** We present the Q-difference plot for IQ-MPC in a problematic scenario (Reacher Hard task in DMControl) where it is affected by an overly powerful discriminator. Although applying a gradient penalty prevents the Q-difference from diverging, it still fails to converge to a value near zero, resulting in a persistently large Q-difference throughout training.

$\mathbb{E}_{(\mathbf{s},\mathbf{a})_{(0:H)}\sim\mathcal{B}_E}Q(\mathbf{z}_t, \mathbf{a}_t)$, and the policy distribution, $\mathbb{E}_{(\mathbf{s},\mathbf{a})_{(0:H)}\sim\mathcal{B}_\pi}Q(\mathbf{z}_t, \mathbf{a}_t)$. While IQ-MPC can mitigate this divergence to some extent through gradient penalty, it does not eliminate the difference entirely, indicating that the policy does not achieve expert-level performance. We show the Q difference plot in a problematic case in Figure 14.

**Long-term Instability** Since we're conducting online imitation learning, we prefer to train a policy that can reach expert-level and maintain stable expert-level performance during further training, which is the long-term training stability. Due to the use of adversarial training, we find it hard for IQ-MPC to maintain stable expert-level performance during extensive long-term online training.

### E.9 IMPROVEMENT OF CONSTRUCTING THE REWARD MODEL ON THE LATENT SPACE

Original RND (Burda et al., 2018) and Random Expert Distillation (Wang et al., 2019) train their reward or bonus models directly on the original observation space. In contrast, we found that constructing the CDRED reward model using the latent representations from a world model yields better empirical performance. This highlights the superior properties of latent representations, which enable more accurate reward estimation. Furthermore, by training a latent dynamics model within this space, the representations become more dynamics-aware, facilitating the construction of a reward model that effectively captures the underlying dynamics.

To validate this, we compared training the CDRED reward model on the original observation space versus the latent space. Our results indicate that while training on the observation space may exhibit slightly suboptimal behavior in low-dimensional settings, it fails entirely in high-dimensional cases due to the challenges of density estimation on raw observations. These findings are illustrated in Figure 15.

## F PROOF OF LEMMA 1

For completeness, we adapt the proof from Yang et al. (2024) to construct the proof of Lemma 1. For a latent state-action pair $(\mathbf{z}, \mathbf{a})$ sampled from a latent state-action distribution $\rho$. We denote the moments of the distribution of random variable $c(\mathbf{z}, \mathbf{a})$ as:

$$\mu_{\bar{\theta}}(\mathbf{z}, \mathbf{a}) = \mathbb{E}\Big[f_{\bar{\theta}_k}(\mathbf{z}, \mathbf{a})\Big] = \frac{1}{K}\sum_{k=0}^{K-1} f_{\bar{\theta}_k}(\mathbf{z}, \mathbf{a}), B_2(\mathbf{z}, \mathbf{a}) = \mathbb{E}\Big[(f_{\bar{\theta}_k}(\mathbf{z}, \mathbf{a}))^2\Big] = \frac{1}{K}\sum_{k=0}^{K-1}(f_{\bar{\theta}_k}(\mathbf{z}, \mathbf{a}))^2,$$

$$B_3(\mathbf{z}, \mathbf{a}) = \mathbb{E}\Big[(f_{\bar{\theta}_k}(\mathbf{z}, \mathbf{a}))^3\Big] = \frac{1}{K}\sum_{k=0}^{K-1}(f_{\bar{\theta}_k}(\mathbf{z}, \mathbf{a}))^3, B_4(\mathbf{z}, \mathbf{a}) = \mathbb{E}\Big[(f_{\bar{\theta}_k}(\mathbf{z}, \mathbf{a}))^4\Big] = \frac{1}{K}\sum_{k=0}^{K-1}(f_{\bar{\theta}_k}(\mathbf{z}, \mathbf{a}))^4.$$

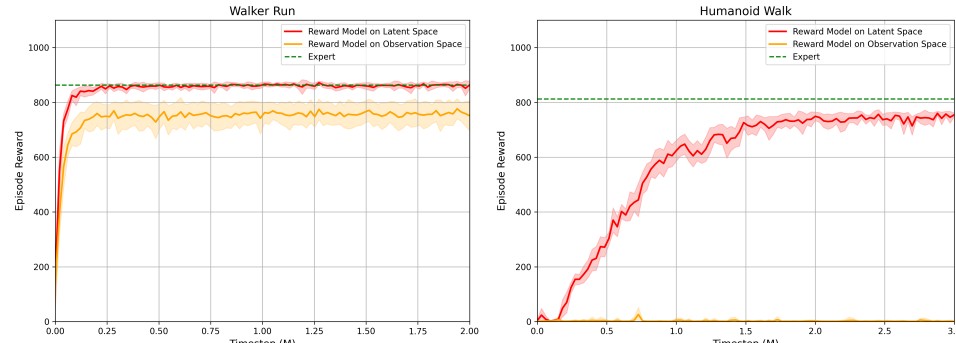

Figure 15: **Effectiveness of the latent space CDRED reward model** We conduct comparative experiments to evaluate the performance of the CDRED reward model when trained on the latent space of the world model versus the original observation space. Our results show that training the CDRED reward model on the latent space yields superior empirical performance.

The calculation for the moments of $f^*(\mathbf{z}, \mathbf{a})$ is as follows:

$$\mathbb{E}[f_*(\mathbf{z}, \mathbf{a})] = \mathbb{E}[\frac{1}{n}\sum_{i=1}^{n} c_i(\mathbf{z}, \mathbf{a})] = \frac{1}{n}\mathbb{E}[\sum_{i=1}^{n} c_i(\mathbf{z}, \mathbf{a})] = \mu_{\bar{\theta}}(\mathbf{z}, \mathbf{a}).$$

$$\mathbb{E}[f_*^2(\mathbf{z}, \mathbf{a})] = \mathbb{E}[(\frac{1}{n}\sum_{i=1}^{n} c_i(\mathbf{z}, \mathbf{a}))^2]$$

$$= \frac{1}{n^2}\mathbb{E}[(\sum_{i=1}^{n} c_i^2(\mathbf{z}, \mathbf{a}) + \sum_{i=1}^{n}\sum_{j\neq i}^{n} c_i(\mathbf{z}, \mathbf{a})c_j(\mathbf{z}, \mathbf{a}))]$$

$$= \frac{1}{n^2}\mathbb{E}[nc^2(\mathbf{z}, \mathbf{a}) + n(n-1)\mu_{\bar{\theta}}^2(\mathbf{z}, \mathbf{a})]$$

$$= \frac{B_2(\mathbf{z}, \mathbf{a})}{n} + \frac{n-1}{n}\mu_{\bar{\theta}}^2(\mathbf{z}, \mathbf{a}).$$

$$\mathbb{E}[f_*^4(\mathbf{z}, \mathbf{a})] = \frac{1}{n^4}\mathbb{E}\left[\sum_{i=1}^{n} c_i(\mathbf{z}, \mathbf{a})\right]^4$$

$$= \frac{1}{n^4}\left(\mathbb{E}\left[\sum_{i=1}^{n} c_i(\mathbf{z}, \mathbf{a})^4\right] + 4\mathbb{E}\left[\sum_{i\neq j} c_i^3(\mathbf{z}, \mathbf{a})c_j(\mathbf{z}, \mathbf{a})\right] + 3\mathbb{E}\left[\sum_{i\neq j} c_i^2(\mathbf{z}, \mathbf{a})c_j^2(\mathbf{z}, \mathbf{a})\right]\right.$$

$$\left. + 6E\left[\sum_{i\neq j\neq k} c_i(\mathbf{z}, \mathbf{a})c_j(\mathbf{z}, \mathbf{a})c_k^2(\mathbf{z}, \mathbf{a})\right] + \mathbb{E}\left[\sum_{i\neq j\neq k\neq l} c_i(\mathbf{z}, \mathbf{a})c_j(\mathbf{z}, \mathbf{a})c_k(\mathbf{z}, \mathbf{a})c_l(\mathbf{z}, \mathbf{a})\right]\right)$$

$$= \frac{nB_4(\mathbf{z}, \mathbf{a}) + 4A_n^2\mu_{\bar{\theta}}(\mathbf{z}, \mathbf{a})B_3(\mathbf{z}, \mathbf{a}) + 3A_n^2 B_2^2(\mathbf{z}, \mathbf{a}) + 6A_n^3\mu_{\bar{\theta}}^2(\mathbf{z}, \mathbf{a})B_2(\mathbf{z}, \mathbf{a}) + A_n^4\mu_{\bar{\theta}}^4(\mathbf{z}, \mathbf{a})}{n^4}.$$

$$(A_n^i = \frac{n!}{(n-i)!})$$

The statistic $y(\mathbf{z}, \mathbf{a})$ is defined as follows in Lemma 1:

$$y(\mathbf{z}, \mathbf{a}) = \frac{f_*^2(\mathbf{z}, \mathbf{a}) - \mu_{\bar{\theta}}^2(\mathbf{z}, \mathbf{a})}{B_2(\mathbf{z}, \mathbf{a}) - \mu_{\bar{\theta}}^2(\mathbf{z}, \mathbf{a})},$$

and its expectation is:

$$\mathbb{E}[y(\mathbf{z}, \mathbf{a})] = \frac{\mathbb{E}[f_*^2(\mathbf{z}, \mathbf{a})] - \mu_{\bar{\theta}}^2(\mathbf{z}, \mathbf{a})}{B_2(\mathbf{z}, \mathbf{a}) - \mu_{\bar{\theta}}^2(\mathbf{z}, \mathbf{a})} = \frac{1}{n}.$$

This implies that the statistic $y(\mathbf{z}, \mathbf{a})$ serves as an unbiased estimator for the reciprocal of the frequency of $(\mathbf{z}, \mathbf{a})$. The variance of $y(\mathbf{z}, \mathbf{a})$ is given by:

$$
\begin{aligned}
Var[y(\mathbf{z}, \mathbf{a})] &= \frac{Var[f_*^2(\mathbf{z}, \mathbf{a})]}{(B_2(\mathbf{z}, \mathbf{a}) - \mu_{\bar{\theta}}^2(\mathbf{z}, \mathbf{a}))^2} \\
&= \frac{\mathbb{E}[f_*^4(\mathbf{z}, \mathbf{a})] - \mathbb{E}^2[f_*^2(\mathbf{z}, \mathbf{a})]}{(B_2(\mathbf{z}, \mathbf{a}) - \mu_{\bar{\theta}}^2(\mathbf{z}, \mathbf{a}))^2} \\
&= \frac{K_1 B_4(\mathbf{z}, \mathbf{a}) + K_2 \mu_{\bar{\theta}}(\mathbf{z}, \mathbf{a}) B_3(\mathbf{z}, \mathbf{a}) + K_3 B_2^2(\mathbf{z}, \mathbf{a}) + K_4 \mu_{\bar{\theta}}^2(\mathbf{z}, \mathbf{a}) B_2(\mathbf{z}, \mathbf{a}) + K_5 \mu_{\bar{\theta}}^4(\mathbf{z}, \mathbf{a})}{n^3 (B_2(\mathbf{z}, \mathbf{a}) - \mu_{\bar{\theta}}^2(\mathbf{z}, \mathbf{a}))^2}
\end{aligned}
$$

where

$$
\begin{aligned}
K_1 &= 1, \quad K_2 = 4n - 4, \quad K_3 = 2n - 3, \\
K_4 &= 4n^2 - 16n + 12, \quad K_5 = -5n^2 + 10n - 6.
\end{aligned}
$$

so we have:

$$
\lim_{n \to \infty} Var[y(\mathbf{z}, \mathbf{a})] = 0.
$$

As $n$ approaches infinity, the variance of the statistic approaches zero, indicating the stability and consistency of $y(\mathbf{z}, \mathbf{a})$.

## USE OF LARGE LANGUAGE MODELS

We used LLMs solely as a writing assistant for minor grammar and phrasing corrections during manuscript preparation. LLMs were not involved in research ideation, experiment design, data analysis, or result interpretation.

