# OpenReview forum: "Coupled Distributional Random Expert Distillation for World Model Online Imitation Learning"
_ICLR.cc/2026/Conference — ICLR 2026 Conference Withdrawn Submission_

### Official Review · Reviewer_pKWQ · 2025-10-16

**Soundness:** 3
**Presentation:** 2
**Contribution:** 3
**Rating:** 4
**Confidence:** 3

**Summary:**

The paper “Coupled Distributional Random Expert Distillation for World Model Online Imitation Learning” proposes a new reward modeling approach for online imitation learning called CDRED. Instead of using unstable adversarial objectives, the method performs coupled density estimation of expert and behavioral state–action distributions in the latent space of a world model, using Random Network Distillation (RND) as the underlying mechanism. By learning these distributions jointly, CDRED aims to improve training stability and exploration during policy learning. Integrated into a decoder-free world model similar to TD-MPC, the approach jointly optimizes latent dynamics, value estimation, and policy learning. Experiments on DMControl, Meta-World, and ManiSkill2 benchmarks demonstrate that CDRED achieves more stable and expert-level performance than adversarial baselines such as IQ-MPC and CFIL, especially in both locomotion and manipulation tasks.

**Strengths:**

1. The paper identifies a well-known issue—training instability in adversarial imitation learning—and proposes a conceptually grounded alternative using density estimation via random network distillation.

2. Incorporating the CDRED reward model into a decoder-free world model (TD-MPC–style) is technically elegant and aligns with recent advances in model-based RL, improving sample efficiency and planning stability.

3. Empirical results across DMControl, Meta-World, and ManiSkill2 convincingly show reduced variance and more stable performance compared to adversarial baselines like IQ-MPC and CFIL.

4. The authors evaluate the method on diverse tasks (locomotion, manipulation, and visual control), providing broad evidence of generality and robustness.

5. The paper provides detailed architecture, hyperparameters, and algorithmic pseudocode (Algorithms 1–2), along with a reproducibility statement and supplementary materials, which support transparency and replication.

6. The proposed stable and model-based imitation learning framework has meaningful implications for robotics applications, where safety and stability are critical.

**Weaknesses:**

1. The proposed Coupled Distributional Random Expert Distillation (CDRED) aims to stabilize imitation learning by coupling expert and behavioral distributions in latent space. However, its innovation over prior methods such as RED (Wang et al., 2019) and CFIL (Freund et al., 2023) remains unclear. The paper should better distinguish CDRED’s theoretical advantages and explain why coupling yields more stable learning.

2. Sections 3.1–3.2 introduce correction terms and balancing factors (α, ζ), but their theoretical rationale and impact on stability are insufficiently explained. The bi-term reward design in Eq. (7) lacks convergence analysis. Adding theoretical discussion or diagnostic plots showing reward evolution would make the stability claim more convincing.

3. Although experiments on DMControl, Meta-World, and ManiSkill2 are extensive, analysis remains qualitative. Claims of “stability” and “better convergence” need quantitative metrics such as variance, confidence intervals, or sensitivity studies on key hyperparameters (ζ, α, K). More ablations would strengthen the empirical validation.

4. Using a TD-MPC–style decoder-free world model is reasonable but not well justified. The claim that latent-space CDRED outperforms observation-space RED/RND lacks direct comparison. Controlled experiments isolating latent-space benefits would clarify CDRED’s necessity and design motivation.

5. The paper is generally clear but includes inconsistent notation (e.g., λ and γ) and dense sentences. Figures 2–3 lack detailed legends explaining predictor and target interactions. Explicitly distinguishing training vs. inference stages and adding a symbol table would improve readability.

6. The paper mentions real-world potential but omits discussion on computational cost or scalability. CDRED’s dual predictors and target ensemble may be expensive. Including runtime analysis and comments on real-robot deployment or partial observability would enhance its practical relevance.

**Questions:**

1. The paper would benefit from a more formal theoretical discussion of why coupled density estimation leads to stable reward learning. Providing convergence insights or variance-boundedness analysis would make the stability claim more convincing.

2. Include ablation results on key hyperparameters (e.g., α, ζ, K) and the effect of the coupling term. Quantifying how these factors influence stability and performance would strengthen empirical rigor.

3. Better justify the choice of a TD-MPC–style world model over alternatives like Dreamer. Directly comparing CDRED in latent versus observation space could isolate and highlight the benefits of latent coupling.

4. Refine notation consistency (λ, γ), expand figure captions to explain training vs. inference processes, and include a concise symbol table in the appendix to improve readability and accessibility for a wider audience.

---

### Official Review · Reviewer_jd9B · 2025-10-17

**Soundness:** 2
**Presentation:** 2
**Contribution:** 2
**Rating:** 2
**Confidence:** 4

**Summary:**

This paper introduces CDRED, a novel framework for stable online imitation learning within world models. Unlike prior adversarial imitation learning methods that often suffer from training instability, CDRED formulates reward estimation as a coupled density estimation problem in the latent space of a world model. Specifically, it jointly learns the expert and behavioral state-action distributions using random network distillation (RND) to construct a consistent and dynamics-aware reward model. This coupled formulation encourages exploration in early training and stabilizes convergence toward expert behaviors later. Extensive experiments on DMControl, Meta-World, and ManiSkill2 benchmarks show that CDRED achieves expert-level performance in both locomotion and manipulation tasks while maintaining remarkable stability across long-term online training.

**Strengths:**

1. The paper identifies a real and important problem: instability in adversarial imitation learning within world models. The replacement of adversarial training with density-based reward modeling is reasonable.
2. The paper evaluates across multiple benchmarks and provides extensive ablations.

**Weaknesses:**

The paper’s overall motivation is not sufficiently clear. Although each component in the proposed framework (e.g., the coupled density estimation, RND-based reward model, and world model integration) appears to be useful on its own—as supported by the ablation studies—the connections among these components are weak and feel somewhat ad hoc. The method seems to be a combination of existing techniques rather than a unified, principled design. In particular, the entire paper is built upon the world model paradigm, yet the proposed algorithm does not appear to leverage the structural advantages of world models beyond simply using the latent space as the feature domain for density estimation. There is no clear justification for why CDRED must be formulated in a world model framework, or how the world model dynamics influence the coupled reward learning. In addition, section 3.3 mostly “follows previous works” without clarifying the design rationale or theoretical necessity of each component.

Another major concern is the absence of theoretical grounding for the proposed reward formulation. Existing adversarial imitation learning methods (e.g., GAIL, ValueDICE, IQ-Learn) are built upon well-established theoretical frameworks such as occupancy measure matching or divergence minimization, which provide convergence guarantees or at least well-defined optimality conditions. In contrast, the reward function in CDRED is largely heuristic, combining several empirically tuned components (e.g., coupled density estimation, bias correction, and scaling factors) without a clear theoretical interpretation. It remains unclear whether optimizing this heuristic reward actually corresponds to minimizing a meaningful divergence between the expert and behavioral policies, or whether it guarantees convergence to an expert-consistent policy under any assumptions.

**Questions:**

1. Is the scaling factor $\zeta$ fixed? If yes, how can it help balance exploration and exploitation, and how to select this parameter?
2.  In Equation (6), the summation over time steps within the horizon H is introduced without explanation. It is unclear whether this temporal aggregation is theoretically motivated, empirically necessary, or merely a heuristic adopted from earlier world-model implementations.

---

### Official Review · Reviewer_NYHw · 2025-10-31

**Soundness:** 3
**Presentation:** 2
**Contribution:** 2
**Rating:** 4
**Confidence:** 3

**Summary:**

This paper presents Coupled Distributional Random Expert Distillation (CDRED), a non-adversarial reward model for online imitation learning (IL) in decoder-free world models. CDRED addresses the instability and discriminator collapse often seen in adversarial IL methods such as GAIL, IQ-Learn, and IQ-MPC.

The method performs latent-space density estimation via Random Network Distillation (RND) using two predictor networks that share an ensemble of fixed random targets—one trained on expert data and the other on behavioral data. The coupled formulation defines rewards from the difference between expert and behavioral prediction errors, balancing imitation and exploration. CDRED is integrated into a TD-MPC2-style world model jointly trained with encoder, dynamics, and value networks. During inference, a Model Predictive Path Integral (MPPI) planner maximizes the cumulative CDRED reward. Experiments on Meta-World, DMControl, and ManiSkill2 show expert-level performance and substantially improved stability compared to IQ-MPC and other baselines.

**Strengths:**

1. **Stable reward formulation**: The coupled RND mechanism jointly estimates expert and behavioral latent distributions, balancing exploitation and exploration. This approach intuitively prevents collapse to sub-optimal expert matching while avoiding the instability of adversarial IL.
2. **Empirical robustness**: CDRED achieves expert-level performance on multiple domains and outperforms IQ-MPC and CFIL on Meta-World and ManiSkill2, while matching their performance on DMControl. Stability metrics and qualitative curves (Figures 4–6) clearly support the authors’ claim of lower gradient variance.
3. **Thorough ablations**: The paper evaluates the effects of latent-space training, coupling, the number of expert trajectories, and the role of world models. The results show that coupling notably improves exploration coverage and stability.

**Weaknesses:**

1. **Clarity and self-containment**: Section 3 introduces the RND correction from Yang et al. (2024) before CDRED’s own contribution, which causes conceptual fragmentation. The derivation of Equations 7 and 8 is abrupt and not self-contained, leaving the reader to infer critical relationships between the bias correction term and the coupled reward. The paper is also hard to follow. The notation is heavy, and the description of the reward construction involves many symbols (e.g., $,\epsilon$, $b$, $\mu_{\bar\theta}$) with little intuition. Important design choices such as the bias correction and the switch between the $L^2$ and variance terms are deferred to the appendix, which makes the main narrative difficult to digest. Readers unfamiliar with RND may struggle to understand the motivation behind the formulation.
2. **Outdated references and missing baselines**: The references stop at 2024, but in 2025 there are multiple works on world‑model‑based imitation learning or RND‑based IL. For example, RND‑DAgger (ICLR 2025) uses RND to trigger expert queries, reducing the need for expert data [1]; Dream to Manipulate (ICLR 2025) uses a compositional world model as a digital twin to augment imitation learning [2]; LUMOS (ICRA 2025) trains a language‑conditioned policy in a world model with intrinsic rewards [3]; and AIME‑NoB (TMLR 2025) integrates online interactions and a surrogate reward to overcome knowledge barriers in IL [4]. None of these papers are cited or compared, and CDRED is therefore evaluated only against older baselines.
3. **Limited novelty relative to concurrent work**: The core idea—using RND‑style density estimation in latent space for IL—closely resembles earlier methods (RED) and concurrent 2025 work. The primary difference is coupling the expert and behavioral predictors, yet the paper does not provide strong theoretical justification or thorough ablations on why the coupling stabilizes learning beyond the empirical toy example and empirical trends.
4. **Incomplete discussion of hyperparameters and sensitivity**: Fixed parameters are used throughout without justification. There is no sensitivity analysis to show how performance depends on these choices.
5. **Missing practical details**: The computational cost of training two predictors and performing MPPI planning is not discussed. It would be useful to know the impact on training time, memory and inference latency.

### *References*
[1] Efficient Active Imitation Learning with Random Network Distillation. ICLR 2025. https://arxiv.org/abs/2411.01894

[2] Dream to Manipulate: Compositional World Models Empowering Robot Imitation Learning with Imagination. ICLR 2025. https://arxiv.org/abs/2412.14957

[3] LUMOS: Language‑Conditioned Imitation Learning with World Models. ICRA 2025. https://arxiv.org/abs/2503.10370

[4] Overcoming Knowledge Barriers: Online Imitation Learning from Visual Observation with Pretrained World Models. TMLR 2025. https://arxiv.org/abs/2404.18896

**Questions:**

1. **Hyperparameter sensitivity**: The constants ζ = 0.8, α = 0.9, and K = 5 are fixed throughout. How sensitive is CDRED to these values? Could different ζ schedules improve the exploration–exploitation balance?
2. **Reward interpretability**: Can the authors visualize or provide statistics on the evolving expert versus behavioral prediction errors? Such insights would clarify the coupling effect.

---

### Official Review · Reviewer_FMgv · 2025-11-03

**Soundness:** 3
**Presentation:** 3
**Contribution:** 3
**Rating:** 6
**Confidence:** 4

**Summary:**

This paper introduces Coupled Distributional Random Expert Distillation (CDRED), a method for online imitation learning in world model frameworks. CDRED addresses instability issues commonly observed in adversarial reward formulations by leveraging a density estimation reward model based on random network distillation (RND) computed in the latent space of the world model. The approach features a coupled density estimation mechanism, jointly estimating both expert and behavioral latent state-action distributions, and integrates bias correction for reward consistency. CDRED is evaluated on a diverse set of tasks, including benchmarks from DMControl, Meta-World, ManiSkill2, and D4RL AntMaze, and compared against both adversarial and non-adversarial imitation learning baselines.

**Strengths:**

1. Principled Formulation for Stable Imitation Learning: CDRED directly tackles the pervasive instability in adversarially trained reward or value formulations (e.g., GAIL, IQ-Learn) by replacing the adversarial component with a distributional density estimation (RND-based) framework, coupled for both expert and behavioral distributions in the latent space. Figure 13 and Figure 14, along with Appendix E.7 (Table 9), clearly quantify the inherent training instability and gradient norm blowups that adversarial methods suffer, demonstrating that CDRED yields significantly smoother and more stable training.

2. Strong Empirical Results Across Domains: The paper provides comprehensive experimental results, including rigorous comparisons in Meta-World (Figure 4, Table 1), DMControl (Figure 5), ManiSkill2 (Table 5), and D4RL AntMaze (Table 8). CDRED consistently outperforms or matches state-of-the-art methods (IQ-MPC, CFIL+SAC, IQL+SAC, HyPE, SAIL) on both low- and high-dimensional tasks, and across domains requiring either state-based or visual observations. Performance on the AntMaze tasks explicitly quantifies the exploration benefits provided by the coupling mechanism.

3. Insightful Analysis via Figures and Ablations: The paper conducts thorough ablation studies, visualized in Figures 7–9 and Tables 6–7, to analyze sensitivity to hyperparameters, coupling, the reward shaping function $g(x)$, model-based vs. model-free setups, and the inclusion of model predictive control. Figure 1 (toy GridWorld) and Figure 3 (empirical convergence) concretely illustrate the benefit of coupled density estimation for exploration and state coverage.

4. Theoretical Care with Bias Correction and Estimation: The use of bias correction (Eq. 5–6, Lemma 1) for distributional consistency, following Yang et al. (2024), is mathematically justified. The presentation is precise, and Equations 4–8 carefully lay out the learning objectives, estimator details, and reward model construction.

**Weaknesses:**

1. Overreliance on Prior Work for Mathematical Innovation: A central technical ingredient—bias correction for distributional RND, estimation of occurrence frequencies, and consistency—is largely imported from Yang et al. (2024). The new contribution is in coupling expert and behavioral distribution estimation in the latent space, yet this coupling is principally an architectural/conceptual extension rather than a fundamentally new algorithm. More serious proof of theoretical improvements (e.g., formal convergence or exploration guarantees) attributable to coupling would add value.

2. Limited Theoretical Analysis of Exploration Effects: While empirical analysis (e.g., Table 8 in AntMaze) is clear, the properties by which coupling theoretically improves sample efficiency or coverage are not formalized. The absence of a formal argument—perhaps in the spirit of showing improved coverage or lower dependence on expert initialization under the coupled scheme—is noticeable given the claims about exploration.

3. Reward Model and Hyperparameter Choices Are Empirically Driven: The construction of the reward (Eq. 7–8) leans on empirically tuned weights ($\zeta$, $\alpha$) and the selection of monotonic activation $g(x)$ (linear vs exponential), but the rationale behind these settings is insufficiently analyzed. Figure 8 (ablation on $g$) and Figure 9 (hyperparameter ablations) demonstrate empirical trends but lack a unifying theory for transferability to other domains/settings.

**Questions:**

1. Could the authors provide more formal theoretical support for the claim that coupled estimators improve exploration and sample efficiency beyond the empirical results (e.g., provable coverage or sample complexity)?

2. How does CDRED perform if the latent space is poorly shaped, or with world models that are weaker (e.g., under-fitting or partial observability)? Is there any robustness analysis to latent model quality?

3. Are there scenarios where adversarial methods outperform CDRED, especially in terms of asymptotic performance rather than stability?

4. Would integrating alternative non-adversarial exploration or distribution matching techniques (e.g., count-based or ensemble exploration in latent space) further boost CDRED or resolve any of its weaknesses?

5. How challenging is it to tune the hyperparameters ($\zeta$, $\alpha$, function $g$)? Does performance degrade rapidly outside optimal values?

---

### Note · Authors · 2025-11-13

I have read and agree with the venue's withdrawal policy on behalf of myself and my co-authors.